# Photochemical conversion of CO to C1 and C2 products mediated by porphyrin rhodium(II) metallo-radical complexes

Hongsen Li[1,6], Boao Han[1,6], Rongyi Wang[1], Wentao Li[2], Wentao Zhang[2], Xuefeng Fu [2], Huayi Fang [3], Fuqiu Ma[4], Zikuan Wang [5] ✉ & Jiajing Zhang [1] ✉

Unimolecular reduction and bimolecular reductive coupling of carbon monoxide (CO) represent important ways to synthesize organic feedstocks. Reductive activation of CO through open-shell pathways, though rare, can help overcome the barriers of many traditional organometallic elementary reactions that are hard to achieve. Herein we successfully achieve the unimolecular reduction of CO to $(TPP)RhCH_2OSiR^1R^2R^3$ (TPP = 5,10,15,20-tetraphenylporphyrin), and the release of products $CH_3OSiR^1R^2R^3$, TEMPO-$CH_2OSiR^1R^2R^3$ and $BrCH_2OSiR^1R^2R^3$ in near-quantitative yield under visible light (420–780 nm), which involves radical formation from Rh-C bond homolysis. Bimolecular CO reductive coupling products, $(TPP)RhCOCH_2OSiR^1R^2R^3$, are then obtained via a radical mechanism. Subsequent treatment with *n*-propylamine, $BrCCl_3$ or TEMPO under thermal or photochemical conditions afford small-molecule bimolecular reductive coupling products. To the best of our knowledge, homogeneous systems which reductively couple CO under photochemical conditions have not been reported before. Here, the use of an open-shell transition metal complex, that delivers more than one kind of small-molecule CO reductive coupling products bearing different functional groups, provides opportunities for useful CO reductive transformations.

Reduction and reductive coupling of CO are important approaches for the synthesis of C1, C2, as well as more complicated organic molecular building blocks. While industrially such transformations are usually done by heterogeneous metal-based catalysts, processes involving homogeneous transition metal compounds are also well known and are especially amenable to detailed mechanistic analysis, compared to their heterogeneous counterparts[1–3]. Like other metal-catalyzed transformations of CO, the reduction and reductive coupling of CO typically proceed via closed-shell, thermal pathways, usually involving the coordination of CO, migratory insertion and reductive elimination as key mechanistic steps[4–13]. The optimization and improvement of such reactions will therefore be eventually hindered by fundamental bottlenecks, such as the conflicts of (1) activating the CO (requiring strong CO coordination) and avoiding catalyst poisoning by the CO (requiring weak CO coordination), as well as (2) high reactivity of the catalyst and swift regeneration of the catalyst via product release.

One promising strategy of activating CO without forming overly stable metal carbonyl complexes is to use open-shell transition metal

[1]School of Pharmacy, Binzhou Medical University, Yantai 264003, China. [2]Beijing National Laboratory for Molecular Sciences, State Key Lab of Rare Earth Materials Chemistry and Applications, College of Chemistry and Molecular Engineering, Peking University, Beijing 100871, China. [3]School of Materials Science and Engineering, Tianjin Key Lab for Rare Earth Materials and Applications, Nankai University, Tianjin 300350, China. [4]Yantai Research Institute of Harbin Engineering University, Yantai 264003, China. [5]Max-Planck-Institut für Kohlenforschung, Kaiser-Wilhelm-Platz 1, Mülheim an der Ruhr 45470, Germany. [6]These authors contributed equally: Hongsen Li, Boao Han. ✉e-mail: zwang@kofo.mpg.de; jiajing_z@bzmc.edu.cn

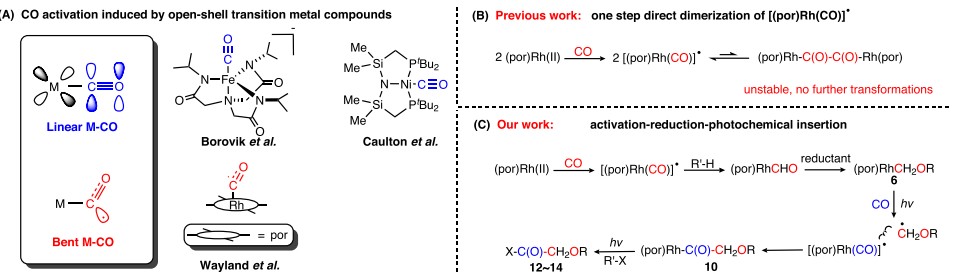

**Fig. 1 | Activation and further transformations of CO mediated by open-shell transition metal complexes. A** CO activation models and examples of CO activation by open-shell transition metal compounds; **B** one step direct dimerization of [(por)Rh(CO)]˙ reported by Wayland; **C** stepwise reductive coupling of CO (this work).

complexes. If the single electron of the metal atom can delocalize into the carbonyl group, this creates a carbonyl complex with formal bond order 2.5 (Fig. 1A), but which is much more reactive than closed-shell carbonyl complexes with a similar extent of back-bonding, due to the radical character of the carbon atom. While it may be tempting to use 3d transition metals to accomplish this task due to their greater tendency to form open-shell complexes, the resulting complexes generally feature linear geometries typical of closed-shell metal carbonyl complexes, like the open-shell Fe-CO and Ni-CO complexes reported by the Borovik[14] and Caulton[15] groups, respectively (Fig. 1A). Furthermore, they are generally not known to exhibit carbon radical character. By contrast, the porphyrin rhodium(II) metallo-radical ([(por)Rh(II)]˙), which features a reactive open-shell 4d transition metal atom, was shown by Wayland et al. to react with CO rapidly[16–19], to generate the porphyrin-rhodium carbonyl radical ([(por)Rh(CO)]˙) with a calculated carbon-oxygen bond length of 1.150 Å and a Rh-C-O angle of 143.7°. This is a very rare activation mode in which the CO ligand bears significant radical character, as evidenced by not only the non-linear geometry (indicative of a sp² hybridized carbon atom) but also a Mulliken spin population of 0.24 at the carbonyl carbon atom[19]. The radical character of [(por)Rh(CO)]˙ has been exploited in many stoichiometric and catalytic transformations of CO[16–20], but most of them involve only one CO molecule. One example is our prior work where [(por)Rh(CO)]˙ reacts with amines to form formamide products, which enabled the catalytic formamide synthesis from amines and CO[19]. Although the dimerization of [(por)Rh(CO)]˙ gives the C-C coupling product α-diketone, this has not been turned into a viable strategy for the synthesis of small-molecule CO coupling products, since the dimer is unstable with respect to dissociation back to the [(por)Rh(CO)]˙ radical (Fig. 1B), due to the intrinsic instability of the C-C bond between two carbonyl groups[21,22].

On the other hand, porphyrin rhodium complexes are known for their ubiquitous photolysis chemistry, where porphyrin Rh-C bonds can be cleanly cleaved homolytically by visible light. Examples are the works by Wayland et al.[20,23] and Collman et al.[24], where porphyrin Rh-C bonds and Rh-H bonds were shown to undergo facile photolysis to yield [(por)Rh(II)]˙ and a carbon/hydrogen radical, both of which are amenable to diverse reactivity. Therefore, one can design reactions that are driven by the formation of strong Rh-C bonds, and then transform the latter to carbon radicals to enjoy their high reactivity and regenerate the porphyrin rhodium catalyst, without suffering from the inevitable compromise between stability and reactivity that one would have to make in thermal reactions. This strategy has enabled the photochemical transformation of CO and many other substrates, as demonstrated by others[20–22,25] and us[19,26–29].

The present work is motivated by the fact that (TPP)RhI (5,10,15,20-tetraphenylporphyrin rhodium iodide), (**1**) and (TPP)RhBr (5,10,15,20-tetraphenylporphyrin rhodium bromide), (**2**) could be reduced by silanes to afford in near-quantitative yield (TPP)RhH[27,30] (**3**) which, in the presence of CO, could be converted to (TPP)RhCHO (**4**) reversibly, via a radical chain mechanism involving the radicals [(TPP)

Rh(II)]˙ and [(TPP)Rh(CO)]˙[31]. In the absence of CO, the unstable **4** could quickly decompose to **3** and CO, which hampers further conversion. Transformation of **4** to a more stable form is therefore essential for the subsequent exploitation of Rh-C photolysis reactivity.

In this work, we design a distinctive strategy (Fig. 1C) to achieve reductive coupling and product release of CO, exploiting the reactivity of [(por)Rh(II)]˙. To stabilize **4** and enable further transformations, we use silane (**5**) to reduce **4** to the Rh-C(sp³) complex (TPP)RhCH₂OSiR¹R²R³ (**6**) by taking advantage of the oxophilicity of silanes[32], which opens up opportunities for subsequent functionalization. By performing the photolysis in the presence of hydrogen sources, BrCCl₃, TEMPO or CO, we are thus able to effect the release of the reduced C1 motif in the forms of CH₃OSiR¹R²R³ (**7**), BrCH₂OSiR¹R²R³ (**8**) or TEMPO-CH₂OSiR¹R²R³ (**9**), or proceed to perform the reductive coupling of CO, giving various C2 products instead. The reductively coupled product is then released from the metal center using a series of strategies, making our system an open-shell CO reductive coupling system where C2 products possessing more than one kind of functional group can be synthesized from the same intermediate.

## Results

### One-pot unimolecular reduction of CO

Wayland's coupling reaction of carbonyl radicals is reversible and gives an equilibrium mixture of [(por)Rh(CO)]˙ and (por)Rh-CO-CO-Rh(por). Though the latter possesses a C-C bond between the two CO moieties, its extreme steric bulk prevented the activation and transformation of the coupling product. We also made corresponding attempts to add reducing agents (hydrogen, silanes), as well as amines and alcohols to a Rh-CO-CO-Rh complex stabilized by a tether group (-O(CH₂)₆O- or a m-xylyl diether)[17,33,34] between the two porphyrin ligands. Unfortunately, subsequent conversion of the Rh-CO-CO-Rh product was not achieved. Therefore, in the present work, we reduced [(TPP)Rh(CO)]˙ before subjecting it to a second molecule of CO, leading to successful generation of small-molecule CO reductive coupling products.

Addition of triethylsilane (HSi(CH₂CH₃)₃, **5a**) to a solution of **1** in C₆D₆ at room temperature resulted in quantitative formation of **3** (δ (ppm): −40.00 (d, Rh-*H*, $^1J_{Rh-H}$ = 40 Hz)) instantly (Fig. 2A. eq. (1)). Pressurizing a solution of the above system with 8 atm of CO in a J. Young valve NMR tube, then placing the mixture at room temperature for 1 day led to a new doublet at δ 3.17 ppm characteristic of (TPP)Rh-C*HO*[20,23,24,35,36] (Fig. 2A. eq. (2)), together with the gradual consumption of **4** through decomposition to H₂ and [(TPP)Rh]₂ (Fig. 2A. eq. (2a)-(2d))[20,23,24,35,36]. Finally, we obtained a mixture of **4, 3** and the poorly soluble [(TPP)Rh]₂. Removing the CO gas from the J. Young valve NMR tube caused significant decrease of the concentration of **4** as evidenced by ¹H NMR. The above experiments proved that **4** and **3** exist in an equilibrium as mentioned in the Introduction, which increases the difficulty of further transformations of **4**. In order to circumvent this difficulty, and in view of the aforementioned lack of reactivity of H₂ and silanes without a catalyst, we decided to reduce **4** with **5a** under the catalysis of B(C₆F₅)₃. After treatment of the above solution with 2 μL of

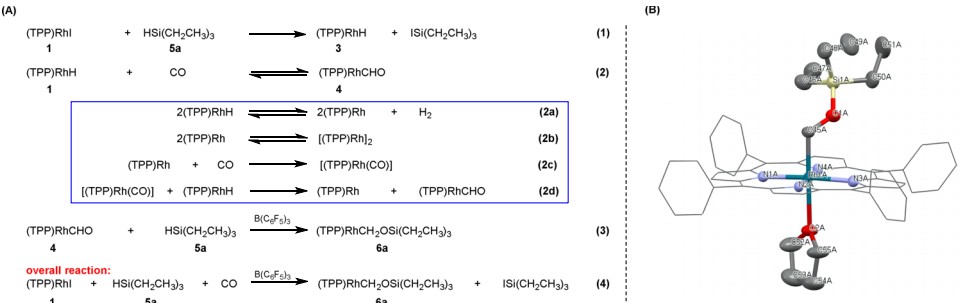

**Fig. 2 | One-pot reaction of CO, 5 and 1 catalyzed by B(C₆F₅)₃. A** Proposed reaction mechanism and the overall reaction. **B** Solid state structure of complex **6a**·THF. H atoms are omitted for clarity. Gray: carbon, yellow: silicon, red: oxygen, light blue: nitrogen, dark blue-green: rhodium.

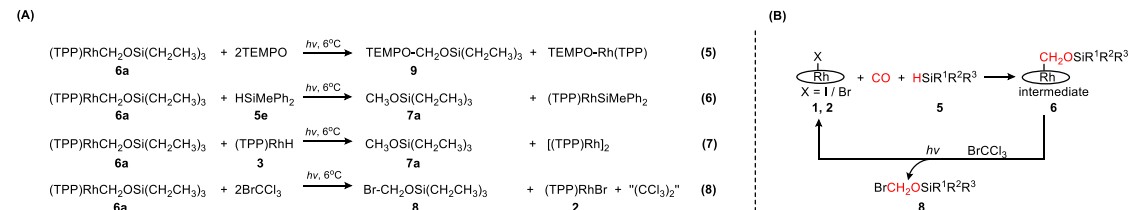

**Fig. 3 | Formation of small-molecule C1 products via the photolysis of 6a. A** Visible light promoted reaction of **5a** with different reagents. **B** Formally catalytic reduction of CO mediated by (TPP)RhX (X = I, Br).

**5a** and 2 mg of B(C₆F₅)₃, we repressurized 1 atm of CO into the J. Young valve NMR tube to afford (TPP)RhCH₂OSi(CH₂CH₃)₃ (**6a**) (Fig. 2A. eq. (3)). The transformation progress was conveniently followed by the appearance of high field ¹H NMR resonances of **6a** (Supplementary Fig. 1) and a rhodium methylene ¹³C NMR signal at δ 52.79 ppm (Supplementary Figs. 2 and 3).

Interestingly, the yield of **6a** was near quantitative, which is much higher than the yield of its precursor, **4**. Evidently, the reduction of **4** drove the equilibrium of the reaction of **3** and CO (Fig. 2A. eq. (2)) to the right side, eventually leading to near complete conversion of **3**. Combining with the fact that **3** is readily obtainable through the reduction of **1** by silanes, we therefore proposed an one-pot strategy, i.e. performing the reduction of **1**, carbonylation of **3**, and the hydrosilylation of **4** in one shot, which allowed us to conveniently scale up the reaction. Mixing the benzene solution containing **3**, B(C₆F₅)₃ and **5a** in a Schlenk flask and then pressurizing 3 atm of CO successfully led to unimolecular reduction of CO in one pot, and stoichiometric production of **6a** was achieved after 12 h (Fig. 2A. eq. (4)). Column chromatography readily afforded the pure product, enabling unambiguous structural characterization through single crystal XRD (Fig. 2B).

To probe the generality of this transformation, the reactivity of a variety of silanes (**5a-5e**) was explored (Supplementary Table 1). Bulkier silanes, such as phenyldimethylsilane (HSiMe₂Ph, **5d**) and methyldiphenylsilane (HSiMePh₂, **5e**), also underwent this transformation smoothly to give the reduction products in moderate to good yields, manifesting the ease of this transformation. Due to the fact that (TPP)RhCH₂OSiR¹R²R³ (**6a-6e**) underwent partial decomposition in the process of column isolation, the isolated yields were about 30 %, lower than those determined by ¹H NMR which were nearly quantitative.

**Release of reduction products**

Since the photolysis of (por)Rh-C bonds yields (por)Rh(II) and carbon radicals, a protocol for the release of the C1 reduction product might be easily envisaged by conducting the photolysis of (TPP)RhCH₂OSiR¹R²R³ (**6a-6e**) in the presence of a radical trap. Thus, reaction of **6a** with excess TEMPO (TEMPO = 2,2,6,6-tetramethylpiperidine-1-oxyl) under visible light irradiation produced TEMPO-Rh(TPP) (Supplementary Fig. 34) and TEMPO-CH₂OSi(CH₂CH₃)₃ (**9**, Supplementary

Fig. 35) (Fig. 3A. eq. (5)). Apart from being a route of product release, this reaction also served as an experimental proof that the photolysis of (TPP)RhCH₂OSiR¹R²R³ proceeded homolytically.

Because the steric hindrance of TEMPO-Rh(TPP) might prevent the exposure of the rhodium center towards external reagents, which are crucial for recovering the initial state of the catalyst, we have sought to use good hydrogen atom donors instead of TEMPO, such as silanes, in the hope that **3** might be formed instead, hence completing a formal catalytic cycle. To examine the feasibility of this route, excess **5e** was added to a solution of **6a** in dry C₆D₆. Surprisingly, after exposure to visible light for 1 h, a 1:1 mixture of (TPP)RhSiMePh₂ and **7a** was observed by ¹H NMR (Fig. 3A. eq. (6)). Compound **7a** was obtained by vacuum distillation as a C₆D₆ solution. A ¹H NMR (Supplementary Fig. 36) single peak of C*H₃*OSi(CH₂CH₃)₃ at δ 3.31 ppm together with a singlet of *C*H₃OSi(CH₂CH₃)₃ at δ 50.08 ppm in ¹³C DEPT NMR (Supplementary Fig. 37) unambiguously confirmed the proposed structure, which was also supported by GC-MS analysis. **6b-6e** also underwent this transformation smoothly to give the corresponding products with near quantitative yields (Supplementary Table 1, Supplementary Figs. 38–45).

In order to elucidate the sources of H and C atoms in compounds **6a** and **7a**, isotopic labeling experiments using either (CH₃CH₂)₃SiD or ¹³CO instead of their natural abundance counterparts were performed, resulting in the formation of (TPP)RhCD₂OSi(CH₂CH₃)₃ (**6f**) and (TPP)Rh¹³CH₂OSi(CH₂CH₃)₃ (**6g**), respectively. The absence of the doublet at −1.73 ppm in the ¹H NMR spectrum of **6f** (Supplementary Fig. 30) and the presence of a strong doublet at 52.41 ppm in the ¹³C NMR spectrum of **6g** (Supplementary Fig. 32) confirmed that the -CH₂OSi(CH₂CH₃)₃ moiety was attributed to the reduction of CO by silanes. Subsequently, addition of (CH₃CH₂)₃SiD (60 % D) to **6a**, followed by irradiation for 2 h at 6 °C, produced DCH₂OSi(CH₂CH₃)₃ (**7f**) (56 % D), which was evidenced by the ¹H NMR triplet peak at 3.28 ppm (Supplementary Fig. 46), confirming the origin of the third methyl proton.

Unfortunately, (TPP)RhSiMePh₂ appeared to be quite inert towards photolysis, unlike its Rh-C homologs: heating at 120 °C or prolonged irradiation under room temperature resulted in no reaction even in the presence of TEMPO as a radical trap. To avoid the generation of (TPP)RhSiMePh₂, we turned our attention to another

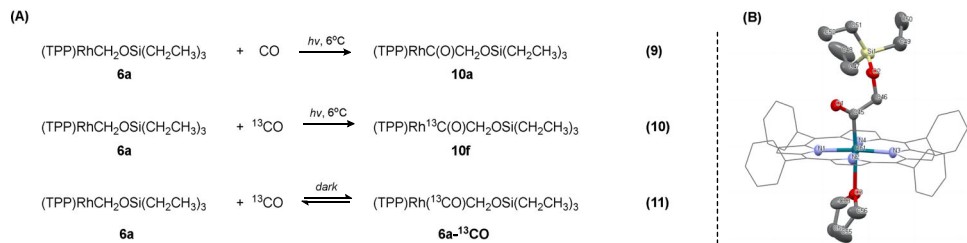

**Fig. 4 | Reaction of the second molecule of CO with 6a. A** Insertion and reversible coordination under light and dark conditions, respectively. **B** Solid state structure of complex **10a**·THF. H atoms are omitted for clarity. Gray: carbon, yellow: silicon, red: oxygen, light blue: nitrogen, dark blue-green: rhodium.

hydrogen source, **3**. Paralleling the reactivity of silanes, the photoreaction of **3** and **6a** was expected to produce [(TPP)Rh]$_2$, whose reaction with H$_2$ is known to produce **3**[30], thus completing a formal catalytic cycle. Exposure a solution of **6a** and **3** in dry C$_6$D$_6$ to visible light for 1 h afforded a dark cloudy orange red solution containing the poorly soluble [(TPP)Rh]$_2$ and (CH$_3$CH$_2$)$_3$SiOCH$_3$ (**7a**, Fig. 3A. eq. (7)), as revealed by $^1$H NMR and ESI-MS. The analogous reactions of **6b-6e** proceeded similarly with high yields (Supplementary Table 1).

Although [(TPP)Rh]$_2$ can react with H$_2$ to complete a formal catalytic cycle (Fig. 2A. eqs. (2a) and (2b)), the reaction is very slow due to the former's poor solubility. It might thus be desirable to generate a precursor to **3** that is more soluble than [(TPP)Rh]$_2$. Bearing this in mind, we utilized excess BrCCl$_3$ instead of hydrogen sources, and obtained BrCH$_2$OSi(CH$_2$CH$_3$)$_3$ (**8**) and **2** (Fig. 3A. eq. (8), Supplementary Figs. 47–49) under visible light irradiation for 2 h. Gratifyingly, removing all the components with low boiling point under vacuum, and adding **5a**, we obtained **3** again with a doublet peak at −40.00 ppm corresponding to the characteristic of Rh-*H* in $^1$H NMR spectrum, suggesting that **2** could be rapidly reduced to **3** by silanes. On the basis of the above observations, a formally catalytic unimolecular reduction of CO was envisioned to proceed through a two-step reaction (Fig. 3B) involving: reduction of CO mediated by the precursor (**1** or **2**) and **5a** in the presence of B(C$_6$F$_5$)$_3$, giving the intermediate **6a**; production of **8** and **2** through photolysis of **6a** in the presence of BrCCl$_3$, completing the formal catalytic cycle.

## Reductive coupling of CO

The facile homolysis of **6a** prompted us to explore the possibility of inserting a second CO molecule into the Rh-C bond, which would constitute a formal reductive coupling of two CO molecules. Pressurizing 8 atm of CO into a dry C$_6$D$_6$ solution containing **6a** in a J. Young valve NMR tube, and then irradiating with visible light at 6 °C over 20 h, led to the CO insertion product (TPP)RhCOCH$_2$OSi(CH$_2$CH$_3$)$_3$ (**10a**) with a yield of 92 % (Fig. 4A. eq. (9)). The existence of a carbonyl group caused the doublet attributable to Rh-C*H*$_2$OSi(CH$_2$CH$_3$)$_3$ (δ (ppm): −1.72 (d), Supplementary Fig. 1) to shift to lower field (δ (ppm): −1.52 (s), Supplementary Fig. 54) in $^1$H NMR spectrum and collapse into a singlet due to attenuated $J_{Rh-H}$ coupling. A doublet with $^1J_{Rh-C}$ = 31.7 Hz at δ 198.00 ppm in $^{13}$C NMR spectrum (Supplementary Fig. 55) and a single carbonyl stretch peak at 1735 cm$^{-1}$ in IR spectrum (Supplementary Fig. 59) further confirmed this assignment. After chromatographic isolation, pure **10a** was obtained and crystallized as its THF adduct, enabling the determination of its structure through XRD, as shown in Fig. 4B and Supplementary Table 6.

To prove the origin of the second carbon atom, an isotope labeling experiment was carried out by replacing CO with $^{13}$CO. Accordingly, the production of (TPP)Rh$^{13}$C(O)CH$_2$OSi(CH$_2$CH$_3$)$_3$ (**10f**, Fig. 4A. eq. (10)) was confirmed by the splitting of the δ −1.52 ppm methylene signal to a doublet ($^2J_{C-H}$ = 2.1 Hz) in $^1$H NMR spectrum (Supplementary Fig. 78) and the appearance of a strong doublet signal ($^1J_{Rh-C}$ = 31.2 Hz) at δ 198.00 ppm in $^{13}$C NMR spectrum (Supplementary

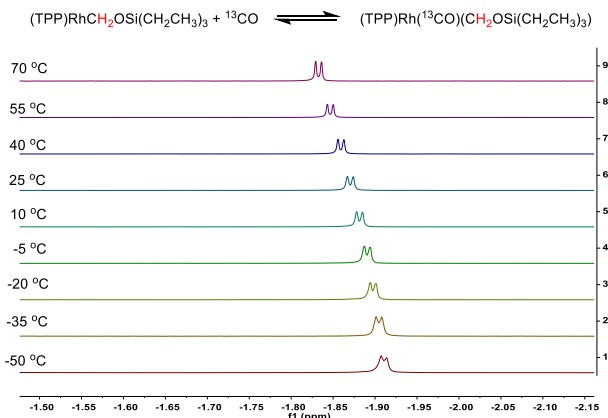

**Fig. 5 | Variable-temperature $^1$H NMR monitoring of the (TPP)Rh($^{13}$CO)(C*H*$_2$OSi(CH$_2$CH$_3$)$_3$) methylene signal in toluene-*d*$^8$.** A progressive shift to the lower fields is seen upon increasing the temperature from −50 °C to 70 °C.

Fig. 79), characteristic of the newly formed carbonyl group. Furthermore, the methylene carbon signal ((TPP)Rh$^{13}$COC*H*$_2$OSi(CH$_2$CH$_3$)$_3$) was split into a dd pattern, with coupling constants $^1J_{C-C}$ = 33.2 Hz and $^2J_{Rh-C}$ = 4.7 Hz. Thus, introducing light irradiation helped the system to overcome the high thermal barrier of the cleavage of the strong (por) Rh-C bond, causing the reductive insertion of the second CO molecule to occur under mild conditions with good substrate universality (Supplementary Table 5).

To probe the exact mechanism of the insertion process, we treated **6a** with $^{13}$CO (**6a**-$^{13}$**CO**) in the dark (Fig. 4A. eq. (11)), which resulted in the shift of the methylene peak from −1.72 ppm to −1.84 ppm, consistent with the *trans* effect of a coordinated CO. Variable-temperature NMR studies showed fluxional behavior. A trend consistent with the *trans* effect of CO was observed in $^1$H NMR, where the rhodium-bound methylene peak ((TPP)Rh($^{13}$CO)C*H*$_2$OSi(CH$_2$CH$_3$)$_3$) shifted to lower fields with increasing temperature as the coordinated CO was gradually detached from the rhodium center (Fig. 5). The $^{13}$C NMR spectra (Supplementary Fig. 53) displayed a singlet, the chemical shift of which (185.28 ppm) being within the typical range of Rh(III) carbonyl complexes[31]. Based on the above observations, we conclude that the intermediate (TPP)Rh(CO)(CH$_2$OSi(CH$_2$CH$_3$)$_3$) (**6a·CO**) was formed, where the CO is coordinated to the *trans* position of the CH$_2$OSi(CH$_2$CH$_3$)$_3$ ligand.

A variety of mechanisms can be proposed for the CO insertion step. The Rh-C bond of **6a** can be photolyzed to yield [(TPP)Rh(II)]$^•$ and $^•$CH$_2$OSi(CH$_2$CH$_3$)$_3$ (as suggested by the TEMPO trapping experiment, Fig. 3A, eq. (5)), both of which might react with CO before being trapped by the other open-shell fragment to yield **10a** (Fig. 6A, a, b). Alternatively, the reaction might proceed through the photolysis of **6a·CO**, forming [(TPP)Rh(CO)]$^{•19}$ without recourse to the bimolecular reaction of [(TPP)Rh(II)]$^•$ and CO (Fig. 6A, c). Computational studies

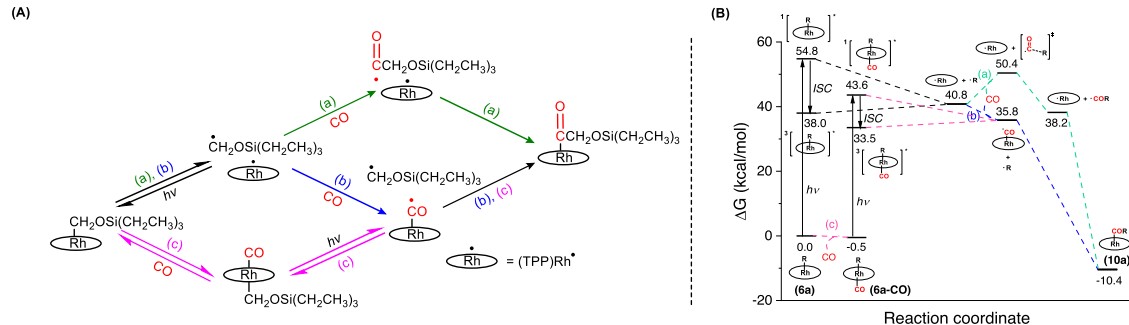

**Fig. 6 | Light-induced CO insertion of (TPP)Rh-CH₂OSi(CH₂CH₃)₃. A** Schematic depiction of possible mechanisms. **B** Gibbs free energy profile of the CO insertion of **6a**. All ground state reactions shown in this figure for which no transition states are given are barrierless. Green, blue and magenta represent the mechanisms (**a**), (**b**) and (**c**), respectively.

performed by ORCA[37] indicate that the reaction of ˙CH₂OSi(CH₂CH₃)₃ with CO in pathway (a) has a barrier of 10.9 kcal/mol, while all other ground state reactions, including the coordination of CO to **6a** and [(TPP)Rh(II)]˙ as well as all radical recombination steps, are barrierless (Fig. 6B). Therefore, pathways (b) and (c) in Fig. 6B are both facile and are probably more important than pathway (a). As shown in Fig. 6B, the equilibrium of **6a** and **6a-CO** is almost thermoneutral and slightly favors **6a-CO**. The result is clearly consistent with the variable-temperature NMR results that both species must have contributed substantially to the equilibrium. Nevertheless, the ground state Rh-C(alkyl) bond dissociation free energy (BDFE) of **6a-CO** is less than that of **6a** by 4.7 kcal/mol, owing to the strong *trans* effect of the CO ligand (the (TPP)Rh(CO)-(CH₂OSi(CH₂CH₃)₃) bond length, 2.050 Å, is 0.030 Å longer than that of **6a**, 2.020 Å; Supplementary Table 8), as well as the spin delocalization effect in [(TPP)Rh(CO)]˙[19] which stabilizes the dissociation product. Thus, the Rh-C bond cleavage in pathway (c) is thermodynamically less unfavorable than the Rh-C bond cleavage in pathway (b). Finally, we note that our NMR experiments failed to detect any coupling product of two carbon radicals, which can be explained by the persistent radical effect, since [(TPP)Rh(II)]˙ and [(TPP)Rh(CO)]˙ are much more stable than the carbon radicals involved in Fig. 6. This may be one of the reasons for the observed high selectivity (> 99 %) of the CO insertion reactions, despite the involvement of highly reactive radicals. An alternative explanation would be that the CO molecule reacts simultaneously with the nascent radicals [(TPP)Rh(II)]˙ and/or ˙CH₂OSi(CH₂CH₃)₃ before they leave the solvent cage; while we failed to locate any stable singlet radical pair between these two species, we found that CO can react with the ˙CH₂OSi(CH₂CH₃)₃ radical in the triplet radical pair, with a somewhat smaller barrier (6.8 kcal/mol) than pathway (a). Due to the triplet multiplicity, however, the reaction is not accompanied by concomitant Rh-C bond formation. Therefore, the reaction of CO with the (TPP)Rh...CH₂OSi(CH₂CH₃)₃ radical pair is more appropriately described as a special case of pathway (a).

To gain further insights into the photolysis steps, we have computed the excited states of **6a**, **10a** and **6a-CO** at their respective ground state equilibrium geometries (Fig. 7). The computed absorptions of **6a** and **10a** match very well with the respective experimental UV-Vis spectra measured in toluene (Fig. 7A). Natural transition orbital (NTO) analyses reveal that the bright states of **6a** and **10a** in the visible region (at 423(415) and 534(522) nm for **6a**(**10a**), commonly known as the Soret and Q bands, respectively) are due to excitations from the two highest ligand π orbitals ($a_{1u}$-like and $a_{2u}$-like; herein we have named the orbitals by the irreducible representations of the $D_{4h}$ group, as is conventionally done) to the two lowest ligand π* orbitals ($e_g$-like), in accord with the well-known Gouterman four-orbital model for porphyrin complexes. The $a_{2u}$-like ligand orbital contains progressively larger contributions from the Rh-C(alkyl) σ orbital upon going

from **6a** to **10a** to **6a-CO**, reaching a maximum of 28 % in the $S_4$ state of **6a-CO** (Fig. 7B); by contrast, while the $a_{1u}$-like and $e_g$-like orbitals contain small contributions from the Rh $d_{xz}$ and $d_{yz}$ orbitals, in none of the cases do they show noticeable contributions from the σ(Rh-C(alkyl)) orbital. This can be explained by symmetry arguments: among the four Gouterman orbitals, only the $a_{2u}$-like orbital has no angular node with respect to the approximate four-fold symmetry axis of the porphyrin ligand, which allows it to mix with the Rh-C(alkyl) σ orbital. Therefore, the Rh-C bond orders of **6a, 10a** and **6a-CO** are reduced by photo-excitation at either the Soret band or the Q band (Fig. 7C), due to the removal of electron density from the $a_{2u}$-like orbital, and therefore from the Rh-C(alkyl) bonding orbital; this is expected to facilitate Rh-C bond cleavage.

The fact that **6a-CO** exhibits the largest σ(Rh-C(alkyl)) component in the $a_{2u}$-like orbital among the three complexes can be rationalized by the orbital interaction between the lone pair orbital of the *trans* CO ligand ($n$(CO)) and the σ(Rh-C(alkyl)) orbital, as revealed by natural bonding orbital (NBO) analysis (Fig. 7C). The interaction results in in-phase and out-of-phase combinations of the two NBOs, the latter being nearly 3 eV higher than the σ(Rh-C(alkyl)) NBO itself. As a result, the out-of-phase NBO combination is energetically very close to the porphyrin $a_{2u}$-like orbital (here approximately represented by the energy expectation value of the $a_{2u}$-like NTO), and thus mixes favorably with the latter. Interestingly, the out-of-phase NBO combination has anti-bonding character between Rh and the CO ligand; therefore, removal of electron density from the $a_{2u}$-like orbital strengthens rather than weakens the Rh-CO bond. This is not only apparent from the Rh-C bond orders (Fig. 7C), but also from the excited state equilibrium geometries and BDFEs. While the Rh-C bond lengths of **6a** and **10a** do not change noticeably upon going from the $S_0$ state to the $S_1$ state, **6a-CO** shows a 0.300 Å increase and a 0.215 Å decrease of the Rh-C(alkyl) and Rh-CO bond lengths, respectively; moreover, the Rh-C(alkyl) BDFE becomes 22.4 kcal/mol lower than the Rh-CO BDFE in the $S_1$ state, even though the former is 32.4 kcal/mol higher than the latter in the $S_0$ state (Supplementary Table 8). Combined with excited state potential energy curve calculations (Supplementary Figs. 96–97), we found that Q band absorption of **6a-CO** leads to selective cleavage of the Rh-C(alkyl) bond but not of the Rh-CO bond, contrary to what would be predicted from ground state BDFEs. A similar but smaller trend is found for the $T_1$ state (Supplementary Table 8).

Apart from CO coordination, the σ(Rh-C(alkyl)) orbital energy can be also raised by a lengthening of the Rh-C(alkyl) bond, due to the narrowing of the σ-σ* gap. Therefore, the $S_1$ states of all three complexes (formed by irradiating the Q band) gradually acquire σ-π* character when the Rh-C(alkyl) bond stretches beyond the excited state equilibrium. Upon further stretching, the high-lying σ*(Rh-C(alkyl)) drops below the porphyrin π* orbitals, resulting in a further transition into ³(σ-σ*) character and therefore complete rupture of the

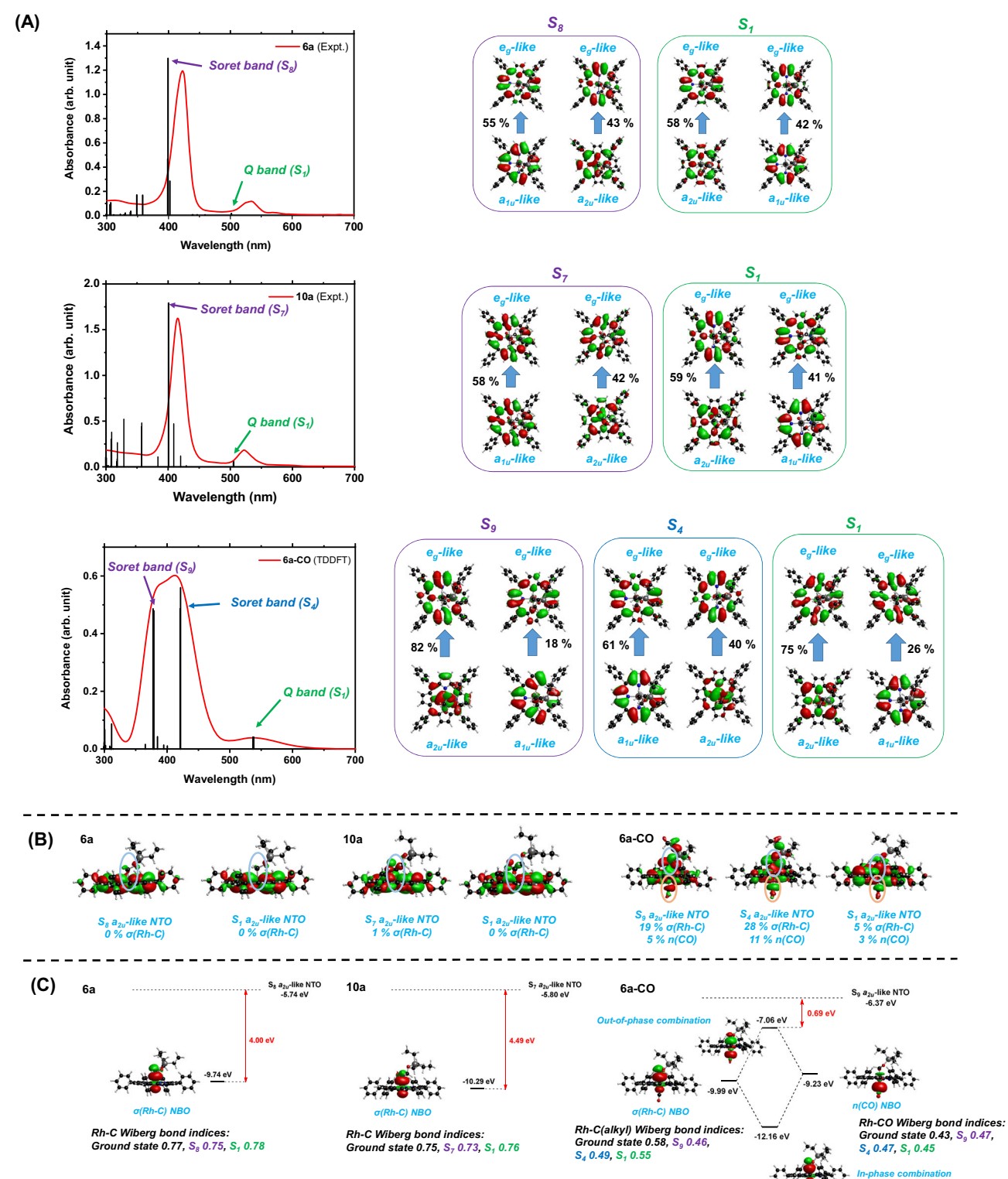

**Fig. 7 | Absorption spectra and excited state compositions of 6a, 10a and 6a-CO. A** UV-Vis spectra of **6a (experimental)**, **10a (experimental)** and **6a-CO (computational)** (red) with computed (TD-PBE0/x2c-TZVPall) absorptions (black) and dominant NTO transitions. The experimental spectra were recorded in toluene. **B** Side views of $a_{2u}$-like NTOs of all excited states, highlighting the contributions of Rh-C bonds. **C** Orbital energy expectation values of the $a_{2u}$-like NTOs and relevant NBOs, and the Rh-C Wiberg bond indices of ground and excited states. The energy splitting due to linear combination of the $n(CO)$ and $\sigma(Rh$-$C(alkyl))$ NBOs is estimated from their off-diagonal Fock matrix element.

Rh-C bond (Supplementary Figs. 94–96). For photolysis via Q band absorption, the transition from π-π* character to σ-σ* character is associated with a barrier of 5.3 (**6a**), 8.4 (**10a**) and 4.5 (**6a-CO**) kcal/mol, respectively, suggesting that the Rh-C(alkyl) photolysis of **6a-CO** may be slightly more kinetically favorable than **6a** (Supplementary Figs. 94–96); this is consistent with the reduced Rh-C(alkyl) bond order in the S$_1$ state due to CO coordination (Fig. 7C). Photolysis via irradiating the Soret band is more complex due to numerous state

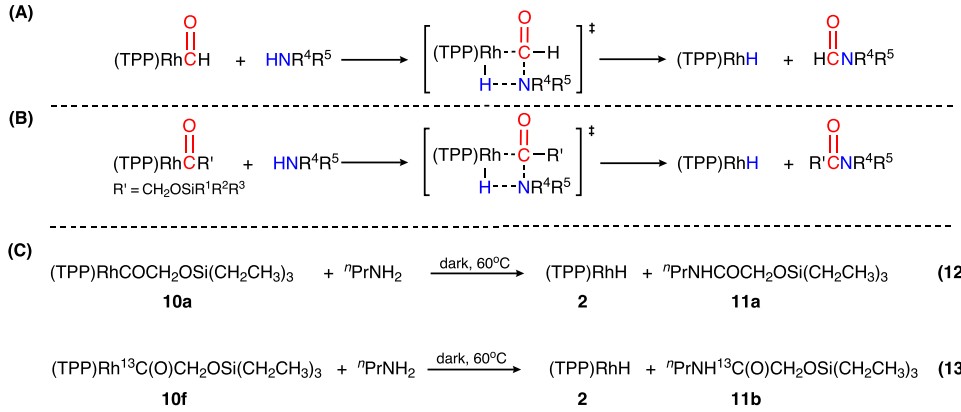

**Fig. 8 | The reaction of (TPP)Rh-COR' with amines. A** Reported mechanism for R' = H[19]. **B** Proposed mechanism for R' = CH$_2$OSiR$^1$R$^2$R$^3$. **C** Reaction of (TPP) RhCOCH$_2$OSi(CH$_2$CH$_3$)$_3$ and (TPP)Rh$^{13}$COCH$_2$OSi(CH$_2$CH$_3$)$_3$, with $^n$PrNH$_2$.

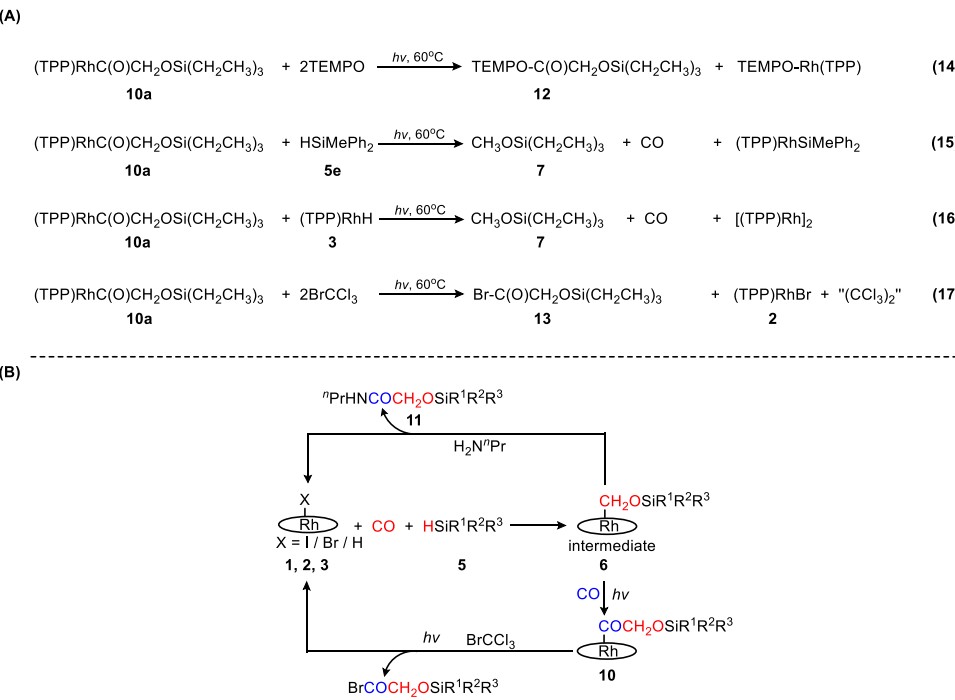

**Fig. 9 | Formation of small-molecule C2 products via 10. A** Visible light promoted reaction of **10** with different reagents. **B** Formally catalytic reductive coupling of CO mediated by (TPP)RhX (X = I, Br).

crossings and may involve the d$_\pi$-π* states, as suggested by the excited state potential energy curves. In general, Soret band photolysis is expected to be roughly equally efficient for the three complexes due to the availability of barrierless or nearly barrierless relaxation pathways that lead to Rh-C homolysis (see discussions of Supplementary Figs. 94–96). To summarize, while for Q band absorption we predict a slight advantage of pathway (c) compared to pathway (b), for Soret band absorption both pathways are about equally viable.

### Release of reductive coupling products

Given the successful synthesis of the reductively coupled products (TPP)RhCOCH$_2$OSiR$^1$R$^2$R$^3$ (**10a-10e**), we turned to the release of the reductively coupled fragments. We have previously reported that **4** could react with $n$-propylamine to produce HCONH$^n$Pr and **3** through a four-membered ring transition state (Fig. 8A)[19]. On account of the similarity in the structures of **4** and **10a-10e**, we proposed that the reactions of **10a-10e** with amines would probably give the respective

amides as well (Fig. 8B). Indeed, adding 0.5 µL of $n$-propylamine to a 1 mg **10a** solution in 300 µL of C$_6$D$_6$, followed by heating at 60 °C for 3 h, resulted in the formation of $^n$PrNHCOCH$_2$OSi(CH$_2$CH$_3$)$_3$ (**11a**) with a yield of 41 %, as shown in Fig. 8C. eq. (12). To aid characterization, the isotopically labeled reactant (TPP)Rh$^{13}$COCH$_2$OSi(CH$_2$CH$_3$)$_3$ was used which produced $^n$PrNH$^{13}$COCH$_2$OSi(CH$_2$CH$_3$)$_3$ (**11b**) (Fig. 8C. eq. (13)) with the signature carbon peak at 169.98 ppm in $^{13}$C NMR, the signature proton peak at 4.08 ppm (d, $^2J_{13C-H}$ = 4.9 Hz; Supplementary Fig. 76) and an ESI-MS spectrum consistent with the incorporation of one $^{13}$C ([M + H]$^+$, 233.17601; Supplementary Fig. 78).

In order to circumvent the low yield of the release product, we designed reaction pathways for more efficient release of the C2 fragment. Inspired by the similarity of the Rh-C bonds of **10a-10e** with those of **6a-6e**, and the computational feasibility of the Rh-C bond photolysis of **10a**, we proposed that **10a-10e** might also undergo Rh-C bond cleavage under photochemical conditions, which would serve as a facile way of product release. Following the

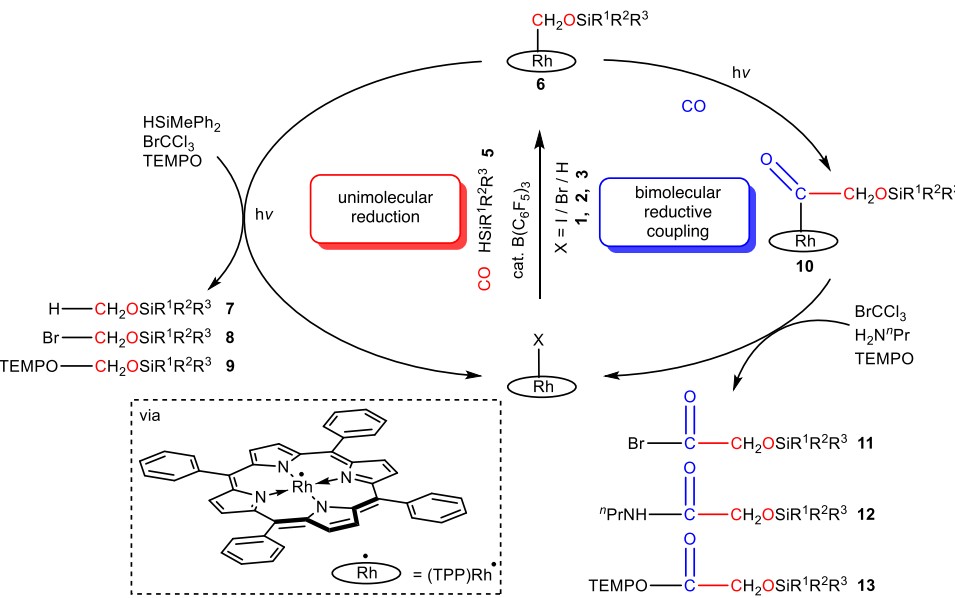

**Fig. 10 | Schematic diagram of unimolecular reduction and bimolecular reductive coupling of CO.** Left (red): single-molecule CO reduction and the corresponding multi-strategy release of reduction products. Right (blue): CO reductive coupling and the corresponding multi-strategy release of reductive coupling products.

previously established protocols of product release of **6a-6e**, TEMPO was first selected as a radical trap. Treating 1 mg of **10a** with excess TEMPO (-10 eq.) and exposing to visible light led to the formation of TEMPO-COC$H_2$OSi$(CH_2CH_3)_3$ (**12a**) (Fig. 9A. eq. (14)) (Supplementary Fig. 80) and TEMPO-Rh(TPP) (Supplementary Fig. 28) as the major products (over 85 % yield as determined by $^1$H NMR spectroscopy, Supplementary Fig. 79). The chemical shift of TEMPO-COC$H_2$OSi$(CH_2CH_3)_3$ (4.22 ppm) appeared downfield from TEMPO-C$H_2$OSi$(CH_2CH_3)_3$ (3.84 ppm), consistent with the presence of a carbonyl group. Next, the hydrogen sources, namely silanes and **3**, were employed as radical traps. Contrary to the high efficacy of the release of the C1 products, however, we failed to trap the carbonyl radical fragments because they are prone to decarbonylation (Fig. 9A. Eqs. (15), (16)). Instead, the C1 product **7a** was observed as the exclusive product, manifesting the hydrogen sources as poorer radical traps compared to TEMPO. To our delight, the trapping reaction using BrCCl$_3$ proceeded smoothly to give the release products **2** and BrCOCH$_2$OSi$(CH_2CH_3)_3$ (**13**) with near quantitative NMR yield, as shown in Fig. 9A. eq. (17). The structure of **13** was verified by $^1$H NMR (Supplementary Fig. 81) and ESI-MS (Supplementary Fig. 82). To our understanding, the production of an acyl halide through the reductive coupling of CO, as demonstrated in this example, has not been previously achieved. The present approach thus significantly expands the potential for subsequent transformations of CO-derived C2 products due to the unique reactivity of the acyl halide functionality. Considering that the release of CO reductive coupling products from metal centers is inherently difficult, the straightforward construction of a C-Br bond during product release, whose bond energy does not offer a significant thermodynamic driving force for the reaction, is truly remarkable. Herein, the utilization of light as an energy source was apparently the key to realizing such reactivity.

A three-step formally catalytic bimolecular reduction of CO was proposed, based on the above observations (Fig. 9B). The steps are the reduction of CO mediated by the precursor (TPP)RhX to give the intermediate **6**; photolysis of Rh-C bond and insertion of the second molecule of CO to produce **10**; production of **13** and **2** through photolysis of **10** in the presence of BrCCl$_3$, completing the formal catalytic cycle.

## Discussion

By integrating light into porphyrin rhodium systems, we illustrate an effective method for formally catalytic unimolecular reduction and bimolecular reductive coupling of CO (Fig. 10), together with the release of C1 and C2 products, respectively (Fig. 3B and 9B). Owing to the photochemistry of porphyrin rhodium complexes, all the reactions proceeded under mild conditions with relatively low CO pressure and without strong reductive conditions. While only small amounts of an unstable formyl complex can be generated slowly from **3** and CO, high yields of hydrosilation products **6a-6e**, important intermediates in subsequent transformations, are generated rapidly from **1** in the presence of silanes and B($C_6F_5$)$_3$ in one pot, taking advantage of the well-known catalytic hydrosilation reaction of aldehydes. The Rh-C bonds of the intermediates undergo homolysis under photochemical conditions to release silyl-substituted formaldehyde acetals, silyl methyl ethers or silyl halomethyl ethers. Incorporation of another CO molecule produces **10a-10e** which, upon employing a variety of strategies, deliver different C2 products bearing various functional groups with high yields.

Traditionally, the activation of CO was typically performed by taking advantage of migratory insertion and reductive elimination reactions, the scope of which are often heavily limited by the nature of the metal and the ligand. Thus, whenever a complex with previously unknown structure is employed in the activation of CO, it is frequently difficult to predict the outcome of a reaction involved in CO activation, due to a lack of knowledge of the chemistry of that particular complex. Therefore, many discoveries on the transformation pathways of CO are serendipitous, and it is difficult to generalize a successful attempt of CO transformation to a systematic approach of CO functionalization that delivers a series of different products. However, our system features a rigid, square planar complex (por)Rh for which migratory insertion and reductive elimination are difficult, if not impossible, owing to the *trans* arrangement of the two vacant sites. As a result, the conventional pathways of CO activation are shut down, leading to a highly selective photochemical approach where the CO-derived axial ligands are readily converted to the respective radicals by light-induced homolysis, with little complications from competing pathways. Since the radicals are already detached from the metal center, their reactivity is perfectly

predictable, and one can apply well-known radical reactions to convert the radicals to a series of products, thus establishing a general protocol for the reductive functionalization of CO. Detailed mechanistic studies and further optimization of reaction conditions towards the design of a real catalytic cycle are underway.

## Data availability

The crystallographic data of complexes **6a**, **6b**, **6d**, **10a** and **10c** generated in this study have been deposited at the Cambridge Crystallographic Data Centre (CCDC) with deposition numbers CCDC 2255347, 2255348, 2255349, 2255350, 2255351. The optimized coordinates of all computationally studied species are provided in Supplementary Data 1. All other experimental and computational data, including the characterization data of all relevant species, are provided in the Supplementary Information file. Additional data related to this paper may be requested from the authors.

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

## Acknowledgements

This work was supported by grants from the National Natural Science Foundation of China (No. 22001020) and Foundation of Binzhou Medical University (No. 2019KYQD23). Z.W. acknowledges generous financial support from the Max Planck Society. We thank Prof. Theodor Agapie (California Institute of Technology) for helpful discussions and Prof. Mingchao Wang (Technische Universität Dresden) for insightful suggestions.

## Author contributions

Conceptualization: X.F., J.Z.; Methodology: B.H., J.Z.; Investigation: J.Z., W.L., Z.W., B.H.; Visualization: H.L., Z.W., J.Z.; Supervision: X.F., J.Z., H.F.; Writing—original draft: J.Z., H.L., R.W.; Writing—review & editing: B.H., J.Z., Z.W., F.M.; Resources: J.Z., X.F.

## Funding

## Competing interests

The authors declare no competing interests.
