## [Peer Review File · Nature Communications]

REVIEWER COMMENTS

Reviewer #1 (Remarks to the Author):

The manuscript details photo-promoted (tpp)Rh-mediated CO transformations, by which the reduction of CO to XCH₂OSiR₃ and XCOCH₂OSiR₃ has been achieved. The later transformation can be viewed as a formal reductive coupling of two COs. In addition to the CO-based transformations, (tpp)Rh-H regeneration from (tpp)Rh halides has also been exercised, which allows the built of the hypothetical catalytic cycles of (tpp)Rh-mediated CO reduction reactions. The key Rh complexes (tpp)RhCH₂OSiR₃ and (tpp)RhCOCH₂OSiR₃ and the organic products formed from their further transformations have been characterized appropriately. In addition, calculations on the thermodynamics of the photo-promoted CO insertion reaction have also been performed, which throw lights on its mechanism. Though these CO transformations are still in stoichiometric version, their radical mechanism is intriguing and should be inspiring for new reaction design for catalytic CO-reduction reactions. The paper should be suitable for publication on Nature Communication Pending the address of the following questions.

- 1) The manuscript is basically well-written. One ignored point is that the compounds in figures lack labels, which makes readers hard to follow. The reviewer thus suggests to add labels for the compounds in figures and schemes. Line 190, complex 6a should be compound6a.
- 2) The paragraph in page 8 notes that (tpp)RhBr can be reduce to 2 by silanes. This reaction might also generate Si-containing byproduct. Is this silyl bromide? Please also comment on the yield of 2 in this transformation.
- 3) Three routes have been proposed for the CO-insertion reaction of 5a and theoretical studies on the thermodynamic gains have been performed on the steps in each route. Route a was thought unlikely since it involves a "small" uphill reaction. This argument is weak as the hill is merely less than 2 kcal/mol above (TPP)Rh + R, and one should also consider the inherent inaccuracy of the calculated energies. Taking this into consideration, the energy differences shown in Figure 6b could not be used as the supporting evidence for a preferred mechanism. If alkyl radicals are generated, one could use radical probes to trapper them, which should give more appealing evidence for the mechanistic proposal.
- 4) A more general question for those photo-prompted CO insertion reactions is the miss of a concerted mechanism that is CO might approach the loosely bound (tpp)Rh...R species formed upon photo-excitation. Calculation study on this mechanism can be added.
- 5) The reaction of 7a with BrCCl₃ was found to generate (tpp)RhBr and BrCOCH₂OSiR₃. This raises questions on the reaction stoichiometry and also the fate of the perhalogenated methane. Please make comments.
- 6) Line 407, Discussion should be Conclusion.

Reviewer #2 (Remarks to the Author):

In the manuscript "Photochemical conversion of CO to C1 and C2 products mediated by porphyrin rhodium(II) metallo-radical complexes," the authors describe the application of well-established rhodium porphyrin photochemistry behavior to chemical derivatization of carbon monoxide. This work is an in-depth organometallic study of the stepwise conversion of CO into varied reduced and/or coupled products. Release of products from the Rh porphyrins is notable due to extensive precedent using Rh porphyrin for small molecule activation but little precedent in non-Rh-bound product formation/release. However, while the experimental results are strong, the topic may not be suitable for the broad audience of Nature Communications because of the narrow focus. Additionally, the chemicals derived from CO do not solve a new need, and there is too much emphasis on a "formal catalytic cycle." Without results demonstrating catalysis, publication of this manuscript may be more appropriate elsewhere.

Technical Comments

1. I suggest that the authors modify their figures showing reaction schemes to label the structures with the appropriate number abbreviation used throughout the text so it is easier to know which species and reaction is being referred to.
2. Wayland previously reported the reductive coupling of CO with a bimetallic Rh porphyrin species as mentioned in the text (J. Am. Chem. Soc. 1997, 119, 34, 7938). Does this precedent influence the proposed pathways in Figure 6? Would there be any utility in a bimetallic strategy for this CO conversion reported in this submitted manuscript?
3. Did the authors attempt a one pot synthesis of bimolecular CO coupled products starting with TPP-Rh-I with silane and CO with irradiation?
4. Do the silane reduced TPP-Rh-CH₂OSiR products need to be isolated before subsequent reaction?
5. The authors mention a "formal catalytic cycle", were any attempts made for truly catalytic formation of reduced CO species? If so, what species were formed?

6. Figure 6a is a little confusing to follow. Can the authors better label the colored arrows and different paths that the reader should follow?

7. For the released products, 6, were isolated yields determined? Table S1 lists the NMR yields, and in the text 6a was isolated by vacuum distillation.

Reviewer #3 (Remarks to the Author):

This contribution by Zhang, Wang and co-workers describes the reductive coupling of CO with hydrogen sources (silanes, amines), TEMPO, and BrCCl₃, respectively, mediated by a rhodium-tetraphenylporphyrin complex. The valorization of CO into useful C1 and C2 molecules is a timely topic of high interest. However, the use of stoichiometric amounts of costly and scarce rhodium for that purpose seems less convenient. This contribution is well focused on the understanding and origin of the reactivity, which is well supported by NMR, HRMS and challenging X-ray diffraction studies of the organometallic complexes evaluated. The claims are well supported by the results obtained and this publication could be publishable after extensive polishing for removing the substantial amounts of experimental details (quantities -mg, microL-, chemical shifts from NMR) from the main text. It seems that the discussed results are just adapted from the experimental part. Compound numbering related to the text will have to be carried out. Overall, this contribution cannot be accepted in its present form. After extensive improving in the scholarship presentation, this contribution could be submitted to a more specialized journal in coordination chemistry. I did not manage to identify the impact that will deserve this publication to be published in a Nature journal.

Minor point: reference 20 should be completed with these two additional references regarding the formation of rhodium-porphyrin hydride species :

J. Am. Chem. Soc. 1991, 113, 5305–5311

J. Am. Chem. Soc. 2000, 122, 11812–11821

Reviewer #4 (Remarks to the Author):

The use of carbon monoxide and other common feedstock materials in chemical synthesis is one of the overarching goals of modern chemistry with the aim of reducing environmental footprint of synthesis. In this work, the authors have managed to utilize carbon monoxide in a radical coupling reaction using readily available porphyrin complexes. It is very surprising that such porphyrin complexes can be successfully employed in radical coupling reactions as the two coupling partners are located on opposing faces of the porphyrin ligand. Despite these challenges, the authors managed the successful realization of such reaction. Overall, even though not being an expert in inorganic-focused, organometallic chemistry, I perceive this work to be of sufficient novelty to warrant publication in Nature Commun.

Before publication, however, several questions arise that should be discussed by the authors:

-It is (for a non-organometallic chemist) sometimes impossible to follow the structure of complexes and/or intermediates. In general, I find only very few compound numbers in schemes, and many in the text, and it is required to flip back and forth in the manuscript main text to grasp, e.g. which complex is 3, or which complex is 6a. Things are further complicated by the nature of 6, which is labeled as CH₃OSiR₁R₂R₃.

-I am missing absorption and emission data of rhodium complexes employed throughout this study. Such data is key to assess, if photochemical processes may occur.

-This UV/Vis data should also be beneficial to better assess reaction pathways discussed in Fig 6. Here, the authors consider that the green pathway (a) is not favorable, although it is only 2 kcal/mol uphill in energy. Comparison of absorption properties of the respective rhodium complex with the computational data (TD-DFT) can provide guidance on potential pathways.

-A more real-world application would be highly desirable. Can the authors show that this transformation can indeed be used for reactions on mmol scale?

-The authors are encouraged to cite: ACIE, 2022, e2022, e202201743, which is a recent review on the photochemical excitation of metal complexes and their use in catalysis.

I would like to express my gratitude to you for your detailed suggestions to improve the manuscript. Thank you very much for taking the time to review this manuscript entitled “Photochemical conversion of CO to C1 and C2 products mediated by porphyrin rhodium(II) metallo-radical complexes” (Manuscript ID: NCOMMS-23-30590). Those comments are valuable and helpful for revising and improving our paper. We have revised the original manuscript according to your comments and made the detailed responses below.

Response to Reviewer 1's Comments

Comments 1: The manuscript is basically well-written. One ignored point is that the compounds in figures lack labels, which makes readers hard to follow. The reviewer thus suggests to add labels for the compounds in figures and schemes. Line 190, complex 6a should be compound 6a.

Response 1: Thank you for your valuable comment. As you suggested, we have now labeled all the compounds.

Labeling: compound **1** for (TPP)RhI, compound **2** for (TPP)RhBr, compound **3** for (TPP)RhH, compound **4** for (TPP)RhCHO, compound **5** for silane, compound **6** for (TPP)RhCH₂OSiR¹R²R³, compound **6-CO** for (TPP)Rh(CO)CH₂OSiR¹R²R³, compound **7** for

$\text{CH}_3\text{OSiR}^1\text{R}^2\text{R}^3$, compound **8** for $\text{BrCH}_2\text{OSiR}^1\text{R}^2\text{R}^3$, compound **9** for
 $\text{TEMPO-CH}_2\text{OSiR}^1\text{R}^2\text{R}^3$, compound **10** for
 $(\text{TPP})\text{RhC}(\text{O})\text{CH}_2\text{OSiR}^1\text{R}^2\text{R}^3$, compound **11** for
 ${}^n\text{PrNHC}(\text{O})\text{CH}_2\text{OSi}(\text{CH}_2\text{CH}_3)_3$, compound **12** for
 $\text{TEMPO-C}(\text{O})\text{CH}_2\text{OSi}(\text{CH}_2\text{CH}_3)_3$, compound **13** for
 $\text{Br-C}(\text{O})\text{CH}_2\text{OSi}(\text{CH}_2\text{CH}_3)_3$.

“Complex 6a” in line 190 has been changed to “compounds **6a** and **7a**”
 marked blue in the main text.

Comments 2: The paragraph in page 8 notes that $(\text{ttp})\text{RhBr}$ can be
 reduce to **2** by silanes. This reaction might also generate Si-containing
 by product. Is this silyl bromide? Please also comment on the yield of
2 in this transformation.

Response 2: Thank you for your insightful comment. We agree with
 you that this reaction would generate silyl bromide as a byproduct. The
 Chan group have reported that the reaction of H-SiR_3 and $(\text{por})\text{RhCl}$
 would produce $(\text{por})\text{RhH}$ and Cl-SiR_3 (*Organometallics*. 2006, 25, 1,
 260-265). In one previous work of our group, we have reported that the
 reaction of silane and $(\text{por})\text{RhI}$ would produce $(\text{por})\text{RhH}$ and I-SiR_3
 (*Organometallics*. 2015, 34, 18, 4507-4514). In the presence of water,
 the so-produced X-SiR_3 compounds will hydrolyze to form HO-SiR_3 or
 $\text{R}_3\text{Si-O-SiR}_3$. We have found the signals of silanol or siloxane in the ${}^1\text{H}$

NMR spectrum (Figure R1 and R2), which is presumably the hydrolysis product of silyl bromide. In Figure R1, the single peak of 8.88 ppm represents pyrrole H on the porphyrin ring of (TPP)RhH, the double quadruple peak in 0.55 ppm and triplet in 0.98 ppm are ethyl hydrogen on silane, and the signal at 3.90 ppm is due to Si-H. Unfortunately, we have no way of distinguishing whether the product is HO-SiR₃ or R₃Si-O-SiR₃. The reason is that their ethyl peaks have similar chemical shifts (Tetrahedron. Lett. 2019, 60, 14, 971-974; J. Am. Chem. Soc. 2000, 122, 48, 12011-12012; Organometallics. 2019, 38, 2, 210-212), leading to overlap. The reduction of (TPP)RhBr or (TPP)RhI to (TPP)RhH had a yield of ca. 100% characterized by NMR. Following your suggestion, we have added “in near-quantitative yield” in the text, line 98.

Figure R1. ^1H NMR spectrum of the reaction mixture of (TPP)RhI and HSiEt_3 in C_6D_6 . The single peak in 8.88 ppm represents pyrrole H in (TPP)RhH; the double quadruple peak in 0.55 ppm and triplet in 0.98 ppm are ethyl hydrogen on silane, and chemical shifts of 3.90 ppm are Si-H.

Comments 3: Three routes have been proposed for the CO-insertion reaction of 5a and theoretical studies on the thermodynamic gains have been performed on the steps in each route. Route a was thought unlikely since it involves a “small” uphill reaction. This argument is weak as the hill is merely less than 2 kcal/mol above (TPP)Rh + R, and one should also consider the inherent inaccuracy of the calculated energies. Taking this into consideration, the energy differences shown in Figure 6b could not be used as the supporting evidence for a preferred mechanism. If alkyl radicals are generated, one could use radical probes to trap them, which should give more appealing evidence for the mechanistic proposal.

Response 3: Thank you for pointing this out. Indeed, with our original data we shouldn't have made the conclusion that pathway (a) is disfavored. However, as our TDDFT calculations necessitate the use of PBE0 for geometry optimization (see our response 4 to reviewer 4), we have reoptimized all structures at the PBE0/x2c-TZVPall level of theory, and at this level of theory we located a transition state for the $\cdot\text{CH}_2\text{OSi}(\text{CH}_2\text{CH}_3)_3 + \text{CO}$ reaction. As PBE0 is also more suitable

for optimizing transition states than BP86 in general due to its larger amount of HF exchange, we believe that our new conclusion is more reliable. The activation energy is small (0.9 kcal/mol), but due to unfavorable entropic contributions, the activation Gibbs free energy is quite large (10.9 kcal/mol; when the reaction proceeds with a nearby [(TPP)Rh(II)][•] radical, the barrier is reduced to 6.8 kcal/mol). We therefore believe that pathway (a) is indeed less important than pathways (b) and (c), based on our new data.

As for mechanistic support using radical trapping, we have already performed TEMPO trapping experiments with **6a**, as shown in Fig. 3A, eq. (5), thereby proving the presence of $\cdot\text{CH}_2\text{OSi}(\text{CH}_2\text{CH}_3)_3$. To make it more obvious to the reader that we have already done such a trapping experiment, we now mention the experiment in the computational section as well (“as suggested by the TEMPO trapping experiment, Fig. 3A, eq. (5)”, lines 290-291).

Comments 4: A more general question for those photo-prompted CO insertion reactions is the miss of a concerted mechanism that is CO might approach the loosely bound (tpp)Rh...R species formed upon photo-excitation. Calculation study on this mechanism can be added.

Response 4: Thank you for your kind comment. We have performed a ground state relaxed potential energy surface scan of the Rh-C bond of

6a (Fig. S94), which shows that there is no stable (TPP)Rh...CH₂OSi(CH₂CH₃)₃ non-covalent complex on the S₀ potential energy surface. Optimizing the structures of the singlet excited states in the dissociation region led to conical intersections, from which the molecule is expected to undergo ultrafast internal conversion to the S₀ state; we were thus unable to locate any (TPP)Rh...CH₂OSi(CH₂CH₃)₃ non-covalent complex on the singlet excited state manifolds either. Triplet states higher than T₁ are similarly expected to undergo ultrafast internal conversion to the T₁ state. Therefore, a loosely bound (TPP)Rh...CH₂OSi(CH₂CH₃)₃ species can only exist on the T₁ state potential energy surface. We indeed located a non-covalently bound, triplet (TPP)Rh...CH₂OSi(CH₂CH₃)₃ structure, and subjected it to a CO molecule; the CO molecule was found to react with the ·CH₂OSi(CH₂CH₃)₃ radical in the complex, but not with the Rh center, since the latter is shielded by the ·CH₂OSi(CH₂CH₃)₃ radical (Fig. R2). The barrier is smaller (6.8 kcal/mol) than if [(TPP)Rh(II)][•] is not present (10.9 kcal/mol), probably owing to the non-covalent interactions between [(TPP)Rh(II)][•] and CO. However, the resulting ·COCH₂OSi(CH₂CH₃)₃ does not spontaneously recombine with the (TPP)Rh part of the complex, because the complex is in a triplet state. Therefore, although CO does react with the (TPP)Rh...CH₂OSi(CH₂CH₃)₃ complex, specifically with the carbon

radical, the reaction cannot be called concerted, and the (TPP)Rh radical acts as a spectator throughout the reaction. We have thus mentioned the reaction of (TPP)Rh...CH₂OSi(CH₂CH₃)₃ with CO as a special case of pathway (a) (lines 315-323).

Fig. R2. Triplet radical pair (TPP)Rh...CH₂OSi(CH₂CH₃)₃ (left), and the transition state (middle) and product (right) of its reaction with CO. The lengths of the forming C-C bond (Å) are shown.

Comments 5: The reaction of 7a with BrCCl₃ was found to generate (tpp)RhBr and BrCOCH₂OSiR₃. This raises questions on the reaction stoichiometry and also the fate of the perhalogenated methane. Please make comments.

Response 5: Thank you for your valuable comment. We believe that each molecule of (TPP)RhCOCH₂OSiEt₃ reacts with two molecules of BrCCl₃, and the CCl₃ group probably becomes hexachloroethane (CCl₃)₂:

We however could not observe the product of the CCl₃ fragment

because it has no peak in the ^1H NMR spectrum. At the same time, hexachloroethane is difficult to ionize in mass spectrometry, so it is difficult to observe by mass spectrometry.

Comments 6: Line 407, Discussion should be Conclusion.

Response 6: Thank you for pointing this out. “Discussion” has been modified to “Conclusion” marked blue.

Response to Reviewer 2's Comments

Comments 1: This work is an in-depth organometallic study of the stepwise conversion of CO into varied reduced and/or coupled products. Release of products from the Rh porphyrins is notable due to extensive precedent using Rh porphyrin for small molecule activation but little precedent in non-Rh-bound product formation/release. However, while the experimental results are strong, the topic may not be suitable for the broad audience of Nature Communications because of the narrow focus. Additionally, the chemicals derived from CO do not solve a new need, and there is too much emphasis on a "formal catalytic cycle." Without results demonstrating catalysis, publication of this manuscript may be more appropriate elsewhere.

Response 1: Thank you for your recognition of this work. We provide a point-by-point answer to your concerns.

(1) Although the study of porphyrin rhodium complex is a specialized field, this does not imply that our manuscript has a "narrow focus." Our work achieves the photochemical reductive coupling of CO molecules, which is itself a topic of broad interest due to the well-known interest in CO conversion and reductive coupling, as well as photochemistry in general, and porphyrin rhodium is merely the tool for us to achieve this reactivity.

(2) The C₂ chemicals synthesized by our method can of course be synthesized by other, known methods. However, the value of the present work does not lie in the synthesis of C₂ products that cannot be synthesized before, but rather that (a) C₂ products (protected hydroxyacetamides and hydroxyacetyl halides) that had to be synthesized by derivatization of other C₂ products can now be synthesized directly from the reductive coupling of CO, instead of necessitating post-modifications after CO reductive coupling; and more importantly (b) the reaction proceeds via a novel mechanism. Therefore, although our reaction is not immediately industrially competitive, we do believe that our work provides useful insights into the design of new, truly catalytic CO reductive coupling systems, including improved versions of our own system.

(3) We emphasized that our system constitutes a “formal catalytic cycle”, because among stoichiometric transformations, those that constitute a formal catalytic cycle are more likely to be converted into a true catalytic cycle than those that do not. There are plenty of CO reductive coupling systems where no known ways are available for regenerating the initial active species, for example due to formation of very strong metal-oxygen bonds. By contrast, in our system the initial porphyrin rhodium complex

((TPP)Rh-X, X = none, Cl, Br, H) can be regenerated swiftly utilizing the light-induced homolysis chemistry of porphyrin rhodium carbon bonds. Our reaction only fails to be a true catalytic reaction because the photolysis condition is currently incompatible with the CO reduction and reductive coupling steps. Nevertheless, reconciling the conditions of two incompatible conditions is always easier than if the condition of one of the reactions (the product release step in the present case) is simply not known. Therefore, our achievement of a formal catalytic cycle is still a non-trivial preliminary result on the way to a true catalytic cycle, and deserves to be reported in a journal targeted at a broad audience.

Moreover, we would like to elaborate further on why we think the present work is of great value as a preliminary study towards a true catalytic cycle. Through mechanistic studies, we would like to demonstrate **the first highlight of this work: “highly activated CO moiety”**. In traditional CO activation strategies, the pivotal philosophy lies in weakening the C≡O triple bond in carbon monoxide via π back-bonding from the coordinated metal center, forming an “activated CO moiety”. The final products are almost universally formed via nucleophilic attack on the weakened C≡O bond (migratory insertion of CO can often be seen as an intramolecular nucleophilic attack on

CO as well), which reduces the scope of substrates eligible for the carbonylation and restricts the design of new CO conversion pathways. However, in our present research, CO was activated via the formation of an active carbonyl-radical-like intermediate $[(\text{por})\text{Rh}(\text{CO})]^\bullet$ (the “activated CO moiety”, see Figure R3) to break the above limitations (J. Am. Chem. Soc. 2018, 140, 21, 6656-6660). Most of the activation of CO is attributed to the partial radical character of the carbonyl carbon atom, which provides a reactivity boost far more than what would be achieved than weakening the $\text{C}\equiv\text{O}$ bond alone, and the radical character provides orthogonal reaction pathways beyond what would be achievable by the traditional electrophilic activation of CO. This unique design makes it possible to achieve CO coupling under mild conditions, i.e. the second highlight of this paper: **CO reductive coupling via a radical reaction mechanism, not only under photochemical conditions but also mediated by late transition metal complexes.**

Figure R3. Calculated spin density on model rhodium porphyrin complexes $[(\text{por})\text{Rh}(\text{II})]^\bullet$

(a) and [(por)Rh(CO)]⁺ (b).

To showcase why our system is more likely to be converted into an industrially useful catalytic reaction than existing stoichiometric approaches, we compare our system with some of the recent and representative systems for stoichiometric CO reductive coupling. One of the important breakthroughs in CO reductive coupling in recent years is the work reported by the Agapie group. They have reported step-by-step CO reductive coupling of Mo compounds by stoichiometric reactions, published in *JACS* (J. Am. Chem. Soc. 2016, 138, 50, 16466-16477; J. Am. Chem. Soc. 2019, 141, 39, 15664-15674) and *Nature* (Nature. 2016, 529, 7584, 72-75). It is worth to mention that harsh reaction conditions were used in Agapie's work, such as the strong reducing agent KC₈. In our work, however, we performed our reactions under room temperature, using the mild reductant silane. It is much more probable to replace silane by an industrially viable reductant (such as H₂) than to replace a strong reductant like KC₈. Zhaomin Hou's research group used Ti, Y and other metal polyhydride complexes, as well as Sm-Li metal complexes, to achieve CO reductive coupling reactions as reported in *JACS* (J. Am. Chem. Soc. 2020, 142, 47, 19889-19894; J. Am. Chem. Soc. 2006, 128, 25, 8124-8125; J. Am. Chem. Soc. 2017, 139, 46, 16967-16973). The formation of a strong metal-oxygen bond is the driving force for the

reductive coupling of CO. However, this also means that there is no known way to cleave the metal-oxygen bonds except by using an external, cheaper reactant that forms an even stronger bond with the oxygen atoms than the transition metals used in the system. Our work is likewise driven by the formation of the strong rhodium-carbon bond, and although the bond is strong enough to prevent thermal product release, we were able to cleave this bond by light irradiation. Although in some of our product release pathways, the resulting carbon radical indeed forms a stronger bond (e.g. a C-O bond) than the starting Rh-C bond, the bond formation is not the driving force for the Rh-C bond cleavage, since the photochemical Rh-C bond cleavage occurred before the C-O bond formation. Therefore, we have solved one of the great challenges of CO reductive coupling (that a strong bond has to be formed to drive the reaction, but it also makes the regeneration of the active catalyst difficult). This not only makes our system one step closer to a true catalytic reaction, but also provides useful insights into improving other CO reductive coupling systems.

Finally, we would like to put our work into the context of the continuous endeavor of the Wayland and Fu groups in CO coupling and transformations. Prof. Wayland and Prof. Fu have published a review in *Science* entitled “Building Molecules with Carbon Monoxide Reductive Coupling” (*Science*. 2006, 311, 5762, 790-791), stressing

that one prominent objective of energy-related research is to develop strategies that use CO as a two-carbon or larger building block to construct the organic compounds needed for fuels and chemical manufacturing. Wayland's group achieved CO coupling by reacting CO with porphyrin rhodium(II) radicals, yielding the Rh-CO-CO-Rh structure. However, steric bulk and instability of the complex prevented further activation and transformation. Now, as a continuation of this work, we used the same porphyrin-rhodium free radical system, except that we reduced the first CO molecule before inserting a second one. This yielded a stronger C-C bond and also helped to reduce the steric hindrance around the formed C-C bond (since the reduced CO moiety no longer carries a porphyrin rhodium group), and therefore led to the successful release of the CO reductive coupling products.

In summary, we believe that our present work possesses sufficient novelty and impact, and is of interest to a broad audience, and we hope the reactivity and mechanism studies in this article will help popularizing the concepts of (1) CO activation via a radical pathway; (2) the use of late transition metal complex systems in CO activation; and (3) the use of photochemical metal-carbon bond homolysis for diverse CO transformations under mild conditions.

Comments 2: I suggest that the authors modify their figures showing

reaction schemes to label the structures with the appropriate number abbreviation used throughout the text so it is easier to know which species and reaction is being referred to.

Response 2: Thank you for your valuable comment. Now all of them have been added. Please see the figures and schemes in the revised version.

(a) CO activation induced by open-shell transition metal compounds

(b) **Previous work:** one step direct dimerization of $[(\text{por})\text{Rh}(\text{CO})]^\bullet$

(c) **Our work:** activation-reduction-photochemical insertion

Fig. 1. Activation and further transformations of CO mediated by open-shell transition metal complexes. (A) CO activation models and examples of CO activation by open-shell transition metal compounds; (B) one step direct dimerization of $[(\text{por})\text{Rh}(\text{CO})]^\bullet$ reported by Wayland; (C) stepwise reductive coupling of CO (this work)

(A)

(B)

Fig. 2. One-pot reaction of CO, 5 and 1 catalyzed by B(C₆F₅)₃. (A) Proposed reaction mechanism and the overall reaction. (B) Solid state structure of complex **6a**·THF. H atoms are omitted for clarity.

(B)

Fig. 3. Formation of small-molecule C1 products via the photolysis of 6a. (A) Visible light promoted reaction of **5a** with different reagents. **(B)** Formally catalytic reduction of CO mediated by (TPP)RhX (X = I, Br).

(A)

(B)

Fig. 4. Reaction of the second molecule of CO with 6a. (A) Insertion and reversible coordination under light and dark conditions, respectively. **(B)** Solid state structure of complex **10a**·THF. H atoms are omitted for clarity.

(C)

Fig. 8. The reaction of (TPP)Rh-COR' with amines. (A) Reported mechanism for R' = H¹⁹. **(B)** Proposed mechanism for R' = CH₂OSiR¹R²R³. **(C)** Reaction of (TPP)RhCOCH₂OSi(CH₂CH₃)₃ and (TPP)Rh¹³COCH₂OSi(CH₂CH₃)₃, with ⁿPrNH₂.

Fig. 9. Formation of small-molecule C2 products via 10. (A) Visible light promoted reaction of **10** with different reagents. (B) Formally catalytic reductive coupling of CO mediated by (TPP)RhX (X = I, Br)

Comments 3: Wayland previously reported the reductive coupling of CO with a bimetallic Rh porphyrin species as mentioned in the text (J. Am. Chem. Soc. 1997, 119, 34, 7938-7944). Does this precedent influence the proposed pathways in Figure 6? Would there be any utility in a bimetallic strategy for this CO conversion reported in this submitted manuscript?

Response 3: Thank you for your comment. We believe that Wayland's report on reductive coupling of CO with a bimetallic Rh porphyrin species (J. Am. Chem. Soc. 1997, 119, 34, 7938) provides some evidence on the presence of pathway (b) in Fig. 6, and does suggest the

presence of the bimetallic (por)Rh-CO-CO-Rh(por) complexes in our reactions, but our experimental findings suggest that (por)Rh-CO-CO-Rh(por) is an off-cycle species.

Wayland et al. characterized the dimer (TMP)Rh-CO-CO-Rh(TMP), bimetallic Rh radicals, and Rh carbonyl radicals, aligning with our [(por)Rh(II)][•] to [(por)Rh(CO)][•] transformation and therefore supporting the presence of pathway (b). Wayland's coupling reaction yields an equilibrium between [(por)Rh(CO)][•] and (por)Rh-CO-CO-Rh(por). Therefore, in all the reactions between [(por)Rh(CO)][•] and an external reagent, the (por)Rh-CO-CO-Rh(por) species is necessarily present, and if only the reactivity of [(por)Rh(CO)][•] is seen (in particular, if there is no product where the CO-CO bond in (por)Rh-CO-CO-Rh(por) is preserved), this can be taken as evidence that (por)Rh-CO-CO-Rh(por) does not participate in the reaction directly. We attempted adding (a) silanes, (b) dihydrogen, (c) amines and (d) alcohols to this equilibrium mixture, but failed to obtain any product where the CO-CO bond was intact. Here, (a) suggests the Rh-CO-CO-Rh complex is not an intermediate in our current work, and (b-d) suggests that it is in general hard to extend our current work and design a CO transformation that incorporates (por)Rh-CO-CO-Rh(por) as an intermediate. Below we will detail these attempts.

- (a) In this work, by using silane, we could realize the direct reduction of CO in one-pot to obtain (por)Rh-CH₂OSiR¹R²R³ (see Fig. 2A in the manuscript). As no reductive coupling products are observed in the dark, we believe (por)Rh-CO-CO-Rh(por) cannot be directly reduced by silane. In fact, the only C-C coupling reaction that we reported here is the reaction of (TPP)Rh-CH₂OSiR¹R²R³ and CO yielding (TPP)Rh-COCH₂OSiR¹R²R³, and ¹³C-labeling experiments have unambiguously proved that the CH₂OSiR¹R²R³ group of the product originates from the CH₂OSiR¹R²R³ group of the reactant, and not from the CO (Fig. 4A, eq. (10)). Therefore, we believe that Wayland's CO coupling reaction is not the reason why we observe C-C bond coupling in our system, despite that both involve C-C bond formation.
- (b) [(por)Rh(CO)]* could react with hydrogen, reversibly forming very small amounts of (por)RhCHO but without any other detectable product, which also shows that (por)Rh-CO-CO-Rh(por) could not react with hydrogen directly.
- (c) In our previous report, [(por)Rh(CO)]* can react with amine to form formamide (J. Am. Chem. Soc. 2018, 140, 21, 6656-6660), which also means that amine cannot react with (por)Rh-CO-CO-Rh(por) on the premise of maintaining the C-C bond (see Page 4 Lines 78-80 in the manuscript). Furthermore, the reactivity has been well

explained by experimental and computational mechanistic studies without invoking the intermediacy of (por)Rh-CO-CO-Rh(por).

(d) Our recent unpublished work also shows that the equilibrium mixture of [(por)Rh(CO)][•] and (por)Rh-CO-CO-Rh(por) reacts with alcohols (ROH) to yield (por)Rh-H and (por)Rh-COOR with very high selectivity, again without detectable products where the CO-CO bond is kept.

Taken together, not only is the conversion of (por)Rh-CO-CO-Rh(por) without C-C bond breaking difficult in the reactions within the present manuscript, but it is also difficult even if we use different reagents than the ones used in the manuscript.

In fact, the present work was initiated as an attempt to overcome the difficulties of designing reductive CO coupling reactions with (por)Rh-CO-CO-Rh(por) as an intermediate. Interestingly, we used the same porphyrin-rhodium free radical system as Wayland did, but we reduced the first CO molecule after it formed the [(TPP)Rh(CO)][•] radical (instead of letting it dimerize), and then inserted the second CO molecule into the reduced Rh-C bond (instead of using a second equivalent of porphyrin rhodium). This way, we were able to obtain a stronger C-C bond that survives the product release, giving small-molecule C₂ products.

Comments 4: Did the authors attempt a one pot synthesis of bimolecular CO coupled products starting with TPP-Rh-I with silane and CO with irradiation? Do the silane reduced TPP-Rh-CH₂OSiR products need to be isolated before subsequent reaction?

Response 4: Thank you for your comment. We have indeed attempted a one pot synthesis, but we found that the final product gives a messy ¹H NMR spectrum that could not be resolved. The silane reduced (TPP)Rh-CH₂OSiR₃ products need to be isolated before subsequent reaction because the catalyst B(C₆F₅)₃ in the system needs to be removed. Control experiments without B(C₆F₅)₃ revealed that B(C₆F₅)₃ was indeed the cause of the phenomenon; photolyzing the system in the presence of both CO and silane, but not B(C₆F₅)₃, yielded (TPP)Rh-SiR₃ cleanly, which is consistent with Fig. 3A, eq. (6). Therefore, one-pot CO reductive coupling is not feasible within this system under the default reaction conditions, although we believe that the problem may be resolved in the future by using another hydrosilylation catalyst (for example one that is not a strong Lewis acid) instead.

Comments 5: The authors mention a “formal catalytic cycle”, were any attempts made for truly catalytic formation of reduced CO species? If so, what species were formed?

Response 5: Thank you for your valuable comment. We have tried to construct a real catalytic cycle, with the reactants TPPRhI, silane, CO and $B(C_6F_5)_3$, and in the absence of light, we have obtained the (TPP)Rh-CH₂OSiR₃ products. Then we tried to directly introduce light to break the Rh-C bond and release the product in situ. However, the results of ¹H NMR spectroscopy data show that when illuminated in the presence of $B(C_6F_5)_3$, the resulting products are too messy to analyze.

We have therefore not yet succeeded with constructing a true catalytic cycle, because of the aforementioned interference of $B(C_6F_5)_3$ in the C-C bond coupling step. In the future, if $B(C_6F_5)_3$ could be replaced by other Lewis acids, or even other non-Lewis acid catalysts, that could catalyze the hydrosilylation of (TPP)RhCHO to (TPP)Rh-CH₂OSiR₃ products, it may be possible to achieve a true catalytic cycle. In the future, we will continue to conduct in-depth exploration and research on catalysts based on your suggestions.

Comments 5: Figure 6a is a little confusing to follow. Can the authors better label the colored arrows and different paths that the reader should follow?

Response 5: Thank you for your kind comment. We have re-labeled and recolored the arrows, so that it is now more clear which path each

arrow belongs to.

Comments 6: For the released products, **6**, were isolated yields determined? Table S1 lists the NMR yields, and in the text **6a** was isolated by vacuum distillation.

Response 6: Thank you for your valuable comment. The NMR yields listed in Table S1 are the products **6**'s yields that we determined at the end of the reaction by adding an external standard. After the determination, the low-boiling fractions were vacuum transferred to a separate vacuum NMR tube. Then the ^1H NMR spectra of **6** were obtained (Figure S1, S8, S15, S19, S26, S30, S31). Although the products have low boiling points, so that they cannot be fully separated from the C_6D_6 solvent, we nevertheless still managed to obtain their pure C_6D_6 solution. Therefore, although we have separated the product in some sense, and obtained rather pure ^1H NMR spectra, the yield that we would obtain by quantifying the product in its pure C_6D_6 solution cannot be called an isolated yield. This differs from the more usual product isolation procedure where the product is first isolated as its solvent-free, pure form, and then dissolved in a deuterated solvent for measuring its NMR spectra.

Response to Reviewer 3's Comments

Comments 1: This contribution by Zhang, Wang and co-workers describes the reductive coupling of CO with hydrogen sources (silanes, amines), TEMPO, and BrCCl₃, respectively, mediated by a rhodium-tetraphenylporphyrin complex. The valorization of CO into useful C1 and C2 molecules is a timely topic of high interest. However, the use of stoichiometric amounts of costly and scarce rhodium for that purpose seems less convenient. This contribution is well focused on the understanding and origin of the reactivity, which is well supported by NMR, HRMS and challenging X-ray diffraction studies of the organometallic complexes evaluated. The claims are well supported by the results obtained and this publication could be publishable after extensive polishing for removing the substantial amounts of experimental details (quantities -mg, microL-, chemical shifts from NMR) from the main text. It seems that the discussed results are just

adapted from the experimental part.

Response 1: Thank you for your recognition and positive comments on our work. We would like to stress that the innovation points of our work are mainly three: (1) the CO activation via an active carbonyl-radical-like intermediate [(por)Rh(CO)]^{*}; (2) the use of late transition metal complexes in CO activation; and (3) the use of photochemical metal-carbon bond homolysis for CO transformations under mild conditions.

As such, the goal of the present work is to provide new insights and ideas into the design of truly catalytic systems, as well as to report our preliminary findings that may lead to a true catalytic reaction from future improvements. The stoichiometric use of rhodium is thus of less concern here, if the present work can inspire the design of (or can be improved into) a truly catalytic reaction in the future. In fact, even within the present work, the porphyrin rhodium can already be recycled and converted to the initial active species nearly quantitatively (although under conditions that are incompatible with the reduction of CO), in contrast to many existing systems where the method for regenerating the active catalyst is not even known. Therefore, we believe that our work will contribute to the future design of industrially competent CO reductive coupling systems, either by improving our own system or by borrowing ideas from our work.

Finally, we would also like to thank you for your valuable comment regarding the experimental details. We have removed the unnecessary experimental details from the main text. We have put all non-essential data into the supporting information, and (when necessary) cited them in the text.

Comments 2: Compound numbering related to the text will have to be carried out.

Response 2: Thank you for pointing this out. Now all compounds for which numbering provides convenience to the reader have been numbered. Please see the figures and schemes in the revised version.

Comments 3: Overall, this contribution cannot be accepted in its present form. After extensive improving in the scholarship presentation, this contribution could be submitted to a more specialized journal in coordination chemistry. I did not manage to identify the impact that will deserve this publication to be published in a Nature journal.

Response 3: Thank you for your comment. We apologize for not being able to highlight the novelty of the article. The novelty of our work can be divided into the following three points:

(1) The novel and rare “highly activated CO moiety” with significant spin density onto the carbonyl ligand is the key highlight, which differs

fundamentally from the traditional CO activation approach that use solely π backbonding to weaken the C \equiv O triple bond.

(2) The use of photoexcitation provides a way out of a long-standing dilemma, i.e. strong bonds like C \equiv O can only be easily activated by very reactive species, but the high reactivity of these species prevents their regeneration (which is necessarily very endergonic). By using light, we can incorporate very endergonic steps in the mechanism (the homolysis of Rh-C bonds, $\Delta G = +30\sim 50$ kcal/mol), while still achieving appreciable reaction rate. This provides a new idea for the activation of inert small molecules in the field of organometallic catalysis.

(3) The use of a rare radical reaction mechanism in CO reductive coupling, providing a versatile strategy for the release of CO reductive coupling products bearing different functional groups. Compared with the widely used rare earth metals and early transition metals, this work is a rare representative of the CO reductive coupling reaction achieved by late transition metals.

Comments 4: Minor point: reference 20 should be completed with these two additional references regarding the formation of rhodium-porphyrin hydride species :

J. Am. Chem. Soc. 1991, 113, 5305–5311

J. Am. Chem. Soc. 2000, 122, 11812–11821

Response 4: Thank you for your valuable comment. We have added two additional references 31 and 32 on porphyrin rhodium hydride and [(por)Rh(II)][•] recommended by the reviewer, next to reference 20 (line 105). At the same time, we have revised the expression in the manuscript to “Indeed, Wayland et al.^{20,21} and Collman et al.²² reported that porphyrin Rh-C bonds and Rh-H bonds could undergo facile photolysis to yield [(por)Rh(II)][•] and a carbon/hydrogen radical, both of which are amendable to diverse reactivity.”

Response to Reviewer 4's Comments

Comments 1: It is (for a non-organometallic chemist) sometimes impossible to follow the structure of complexes and/or intermediates. In general, I find only very few compound numbers in schemes, and many in the text, and it is required to flip back and forth in the manuscript main text to grasp, e.g. which complex is 3, or which complex is 6a. Things are further complicated by the nature of 6, which is labeled as CH₃OSiR¹R²R³.

Response 1: Thank you for your valuable comment. Now all of them have been added. Please see the figures and schemes in the revised version.

Comments 2: I am missing absorption and emission data of rhodium complexes employed throughout this study. Such data is key to assess, if photochemical processes may occur.

Response 2: Thank you for your valuable comment. We apologized for not having been able to present the above spectra in the previous version of the manuscript. We have supplemented the UV-Vis spectra of (TPP)RhI, (TPP)RhCH₂OSiR¹R²R³ and (TPP)RhC(O)CH₂OSiR¹R²R³ in the supporting information (Figure S4, S11, S22, S50, S56). Toluene was used as the solvent, and the maximum absorption wavelengths were all around 410 nm and 520 nm,

Detailed information can be found in the figures below.

Fig. S50. UV-Vis spectrum of (TPP)RhI (**1**) in toluene. UV-Vis: λ_{abs} (nm) (toluene) 374, 430, 540, 570.

Fig. S4. UV-Vis spectrum of (TPP)RhCH₂OSiEt₃ (**6a**) in toluene. UV-Vis: λ_{abs} (nm) (toluene) 422, 533, 570.

Fig. S11. UV-Vis spectrum of (TPP)RhCH₂OSiMe₂Et (**6b**) in toluene. UV-Vis: λ_{abs} (nm) (toluene) 414, 521, 571.

Fig. S22. UV-Vis spectrum of (TPP)RhCH₂OSiMe₂Ph (**6d**) in toluene. UV-Vis: λ_{abs} (nm) (toluene) 414, 521, 549.

Fig. S56. UV-Vis spectrum of (TPP)RhCOCH₂OSiEt₃ (**10a**) in toluene. UV-Vis: λ_{abs} (nm) (toluene) 415, 521, 574.

We however could not see significant fluorescence and phosphorescence experimentally with an excitation wavelength around 520 nm, perhaps due to low quantum yields; as shown in Figure S5, S12, S23, S51 and S57, the luminescence intensities of all tested porphyrin rhodium complexes are negligible compared to the free TPP ligand, which is known to fluoresce with a modest quantum yield $\Phi_{\text{F}} = 0.090$ (Photochemistry Reviews. 2021, 46, 100401). However, the phosphorescence emission wavelength of (TPP)RhI is known at 77 K to be 735 nm (in 2MeTHF), in which condition no fluorescence was detected; the emission properties of other porphyrin rhodium complexes are generally similar (Photochem Photobiol. 2006, 82, 1, 171-176).

Fig. S51. Luminescence spectrum of (TPP)RhI (**1**) in toluene, (TPP)H₂ for reference.

Fig. S5. Luminescence spectrum of (TPP)RhCH₂OSiEt₃ (**6a**) in toluene, (TPP)H₂ for reference.

Fig. S12. Luminescence spectrum of (TPP)RhCH₂OSiMe₂Et (**6b**) in toluene, (TPP)H₂ for reference.

Fig. S23. Luminescence spectrum of (TPP)RhCH₂OSiMe₂Ph (**6d**) in toluene, (TPP)H₂ for reference.

Fig. S57. Luminescence spectrum of (TPP)RhCOCH₂OSiEt₃ (**10a**) in toluene, (TPP)H₂ for reference.

The experimental maximum absorption wavelengths of **6a** (422, 533 nm) correspond to energies of 67.8 and 53.6 kcal/mol, respectively; those of **10a** (415, 521 nm) correspond to 68.9 and 54.9 kcal/mol, respectively. As such, they are larger than the calculated bond dissociation free energies of the Rh-C bonds of **6a** and **10a**, which are respectively 35.4 and 46.9 kcal/mol at the level of theory used in the previous version of the manuscript, and 38.6 and 50.2 kcal/mol at the new level of theory used in the present revised manuscript (see our reply to comment 3 for the reason why we have to change our level of theory). Therefore, thermodynamically the absorption of a photon suffices to lead to Rh-C bond cleavage of both **6a** and **10a**. In our reply to comment 3 we will provide further computational evidences that the

Rh-C bond cleavage reactions of **6a** and **10a**, and even more so **6a-CO**, are also kinetically feasible.

Comments 3: This UV/Vis data should also be beneficial to better assess reaction pathways discussed in Fig 6. Here, the authors consider that the green pathway (a) is not favorable, although it is only 2 kcal/mol uphill in energy. Comparison of absorption properties of the respective rhodium complex with the computational data (TD-DFT) can provide guidance on potential pathways.

Response 3: Thank you for your comment. As we explained in response 3 of Reviewer 1's comments, we now acknowledge that the 2 kcal/mol barrier is not a sufficient evidence that pathway (a) is unfavorable, but after improving our level of theory, we found a 10.9 kcal/mol free energy barrier for this step (or 6.8 kcal/mol when a nearby [(TPP)Rh(II)][•] radical is present), which does suggest that pathway (a) is not favored.

We have also added extensive TDDFT calculations in the revised manuscript (pages 12-14). Our calculations show that **6a-CO** has a much weakened Rh-C(alkyl) bond in the S₁ excited state compared to **6a**, in terms of not only bond length and bond order, but also cleavage barrier. This is traced to the strong *trans* effect of the CO ligand, which pushes up the $\sigma(\text{Rh-C(alkyl)})$ orbital energy and increases the

$\sigma(\text{Rh-C(alkyl)}) \rightarrow \pi^*$ character of the lowest excited states, thereby weakening the Rh-C(alkyl) bond. This effect is expected to increase the relative contribution of pathway (c) compared to pathway (b), when the Q band is excited; when the Soret band is excited, however, the difference between the contributions of pathways (b) and (c) is probably smaller.

As the BP86 functional used in the original manuscript is not suitable for optimizing the excited state geometries (due to the presence of partial charge transfer character in the lowest excited states), we have performed the TDDFT calculations with the PBE0 functional. To allow the comparison of ground state reaction free energies with the excited state results, we have recalculated all ground state free energies with the PBE0 functional as well. This has led to some changes in the free energies shown in Fig. 6B (mostly within 5 kcal/mol, with the exception of the $\cdot\text{CH}_2\text{OSi}(\text{CH}_2\text{CH}_3)_3 + \text{CO}$ reaction barrier), compared to the original manuscript. However, no qualitative conclusions are affected, except that pathway (a) now has a barrier. We have also plotted the excited state free energies on Fig. 6B to aid their comparison with the existing free energy values.

Comments 4: A more real-world application would be highly desirable.

Can the authors show that this transformation can indeed be used for

reactions on mmol scale?

Response 4: Thank you for your kind comment. The first step of our formally catalytic reaction, i.e. synthesis of **6**, can be scaled up to the 40 μmol scale in Schlenk flasks, as we have already reported in our supporting information, page 2 “Synthesis and characterization of (TPP)RhCH₂OSiR₁R₂R₃: 36 mg of (TPP)RhI was dissolved in a 25 mL Schlenk flask with 2.0 mL of toluene, 20 μL of silane and 3 mg of B(C₆F₅)₃.”; we have not tried larger scales, but we do not foresee any difficulty given that the reaction is homogeneous (i.e. without significant mass transfer problems) and occurs under dark. We have tried to do the following photochemical steps at a 40 μmol scale in Schlenk flasks, but they failed to give appreciable yields, because (1) Schlenk flasks are incompatible with the CO pressure (8 atm) used in our small-scale studies, which forces us to use smaller pressures of CO (within 3 atm), and while e.g. autoclaves provide the required pressure resistance, they are not transparent to the incident light; (2) Schlenk flasks have much longer light paths than J. Young NMR tubes, making the light intensity distribution inhomogeneous.

Comments 5: The authors are encouraged to cite: ACIE, 2022, e2022, e202201743, which is a recent review on the photochemical excitation of metal complexes and their use in catalysis.

Response 5: Thank you for your useful comment. We have added this

article in reference 24.

REVIEWERS' COMMENTS

Reviewer #1 (Remarks to the Author):

The authors addressed my concerns appropriately. I support the publication of this manuscript in Nature Commun.

Reviewer #2 (Remarks to the Author):

I thank the writers for making the changes that most reviewers requested, especially by improving the readability of the figures and text so that it is easier to follow each series of reactions.

1. The authors mention that the "photolysis condition is currently incompatible with the CO reduction and reductive coupling steps." Have you attempted performing the reaction where the light source is alternately turned on and off to allow both light and dark reactions to occur in one pot?

2. The authors emphasize the importance of the "activated CO moiety" being a carbonyl radical like intermediate as a new mode of activation that opens new reactivity. However, Wayland around demonstrated this activation through the coupling of CO using Rh porphyrins, the only difference being his group did not accomplish product release. Nonetheless, I disagree with emphasizing this as a unique CO activation mode.

3. Similarly, the authors put this work into context of CO coupling reaction with Rh porphyrins and cite a 2006 Science article, which is nearly 20 year old.

4. Obtaining more traditional isolated yields rather than NMR yields would be beneficial.

5. I am very impressed with this work and have been involved with Rh porphyrin chemistry in the past, so these results are exciting, but may be better suited for a different journal.

Reviewer #4 (Remarks to the Author):

The revised manuscript provides suitable response to most of the reviewer comments and main arguments are now only on final polishing of the manuscript.

-Transitions shown in Figure 7a are barely legible and should be increased in size or moved to SI.

-The results on the challenges for scale-up should be mentioned in the main manuscript text.

I am deeply appreciative of the continued attention and insightful comments you have provided on our manuscript titled “Photochemical conversion of CO to C1 and C2 products mediated by porphyrin rhodium(II) metallo-radical complexes” (Manuscript ID: NCOMMS-23-30590). Your previous feedback has been invaluable in refining our work, and we have made significant efforts to incorporate your suggestions into a revised version of the manuscript.

We are now submitting this revised manuscript for your further consideration. Below, we have outlined our responses to each of your comments, detailing the changes we have made and explaining how they address your feedback. We hope that you find the revised version of the manuscript satisfactory for publication.

Response to Reviewer 2's Comments

Comments 1: The authors mention that the "photolysis condition is currently incompatible with the CO reduction and reductive coupling steps." Have you attempted performing the reaction where the light source is alternately turned on and off to allow both light and dark reactions to occur in one pot?

Response 1: Thank you for your query. Regarding the photolysis condition and its incompatibility with the CO reduction and reductive

coupling steps, we have indeed attempted to address this issue by alternating the light source on and off in an attempt to facilitate both light and dark reactions in a one-pot synthesis. However, our experiments with alternating light exposure for half an hour followed by dark conditions for another half an hour have not yielded satisfactory results, as the products still exhibit a messy ^1H NMR spectrum, similar to what we observed in our initial one-pot synthesis attempts. One possible reason may be that the side reactions during photolysis in the presence of $\text{B}(\text{C}_6\text{F}_5)_3$ are irreversible, such that their adverse effects are not eliminated by simply turning off the light. This suggests that the alternating photolysis conditions are not sufficient to resolve the incompatibility between the photolysis, CO reduction, and reductive coupling steps within our current system.

We believe that further optimization of the reaction conditions, including the selection of a more suitable hydrosilylation catalyst that is not a strong Lewis acid, may be necessary to achieve a successful one-pot synthesis that incorporates both light and dark reactions. We are currently exploring these possibilities and hope to make progress in the future.

Comments 2: The authors emphasize the importance of the "activated CO moiety" being a carbonyl radical like intermediate as a new mode of

activation that opens new reactivity. However, Wayland around demonstrated this activation through the coupling of CO using Rh porphyrins, the only difference being his group did not accomplish product release. Nonetheless, I disagree with emphasizing this as a unique CO activation mode.

Response 2: Thank you for your thoughtful comment regarding the novelty of the "activated CO moiety" as a carbonyl radical-like intermediate in our work.

I agree with you that while the "activated CO moiety" is indeed crucial in our study, it is not the first instance of such an activation mode. However, as you have noted, our work differs significantly from previous approaches, such as those by Wayland using porphyrin rhodium. In particular, we have achieved not only the activation of CO but also its successful reductive coupling and product release, which had not been accomplished previously.

Our approach involves the reaction of an alkyl radical with the rhodium carbonyl radical to achieve CO coupling, rather than two rhodium carbonyl radicals as used by Wayland (from which no known ways exist for releasing the CO coupling product without breaking the nascent C-C bond). This difference not only offers a novel reactivity pathway but also facilitates product release, making it a unique design for CO coupling reactions.

Based on your feedback, we have revised the manuscript to clarify that while the "activated CO moiety" is important in our work, we do not claim it as a unique mode of CO activation. Instead, we emphasize the novelty of our approach in achieving successful CO reductive coupling and product release.

Comments 3: Similarly, the authors put this work into context of CO coupling reaction with Rh porphyrins and cite a 2006 Science article, which is nearly 20 year old.

Response 3: Thank you for your comment. It is worth noting that while the 2006 Science article cited by the authors provided an early insight into the Rh porphyrin-mediated CO coupling reaction through radical-induced pathways, the field has evolved and remained active over the years. Recent advancements, including those reported by Hou, Kays, Agapie, and others (cited in references 6-9 of the manuscript), demonstrate the ongoing interest and progress in CO reduction and coupling reactions. Thus, it is the much more recent references 6-9 that we cited, rather than the 2006 Science paper, that serve to put our work into the context of recent advancements of the field.

Importantly, our 2018 publication (cited in reference 19 of the manuscript) highlights the pivotal role of Rh porphyrin carbonyl radicals in catalyzing CO activation and transformation, representing a

significant milestone in this research area. The fact that it took almost two decades for new breakthroughs reported in the present work (porphyrin Rh radical-mediated CO reduction coupling and product release) to emerge further underscores the novelty and excitement surrounding these recent developments.

We appreciate the reviewer's observation and believe that our work, placed in the context of this evolving field, offers valuable insights into the mechanisms and potential applications of Rh porphyrin-based catalysts for CO coupling reactions. We are committed to citing the most relevant and recent literature to provide the reader with a comprehensive understanding of the current state of the art in this research area.

Comments 4: Obtaining more traditional isolated yields rather than NMR yields would be beneficial.

Response 4: Thank you for your comments on our work. Regarding your suggestion to obtain traditional isolated yields, we fully appreciate the importance of such data. However, in our case, due to the scale of the reaction (the reactions were carried out in vacuum-adapted NMR tubes), the products were obtained in very small amounts as a dilute solution, approximately 1 mg in 300 μL of C_6D_6 . Even when the reaction is scaled up in Schlenk flasks, the yield

remains limited to around 6 mg in 2.0 mL of toluene. As the products are volatile, simply removing the benzene/toluene solvent under vacuum will result in severe to complete loss of the product, but the small amount of products makes distillation impractical for separation and collection. As a result, we have resorted to using NMR with external standards for quantification, which has proven to be more accurate than weighing such small quantities of product (which would be very inaccurate even if the products were not volatile). We believe this approach allows us to obtain the most accurate yield data under our current experimental conditions.

Comments 5: I am very impressed with this work and have been involved with Rh porphyrin chemistry in the past, so these results are exciting, but may be better suited for a different journal.

Response 5: Thank you for your comment. I would like to elaborate on the key highlights of our research to further justify its novelty and significance.

Firstly, our study represents a significant breakthrough in the activation of carbon monoxide (CO). Instead of relying solely on traditional methods that involve weakening the $C \equiv O$ bond through π -backbonding from the metal center, we have successfully demonstrated an unconventional activation mechanism by forming an

active carbonyl radical-like intermediate, [(por)Rh(CO)][•]. This “highly activated CO moiety”, with its partial radical character at the carbonyl carbon atom, offers reactivity beyond that achievable through mere weakening of the C≡O bond. This novel activation strategy not only widens the substrate scope for carbonylation reactions but also opens up orthogonal reaction pathways beyond what is achievable by traditional electrophilic CO activation.

Secondly, our system enables the reduction and coupling of CO under mild conditions, mediated by late transition metal complexes. This achievement breaks the limitation of traditional methods that often use early transition metals and therefore require harsh reaction conditions to break the stable metal-oxygen bonds, making our approach more practical for potential industrial applications. Furthermore, by using light as the energy source, efficient CO coupling was achieved at room temperature. Our system therefore manifests multiple advantages over existing stoichiometric methods, which point to its promising potential in future practical applications.

In summary, our research offers a novel approach to CO activation and reductive coupling, which not only addresses the limitations of traditional methods but also opens up new possibilities for CO utilization in catalytic reactions. We believe that these key highlights, along with the novelty and significance of our findings, justify the

publication of our work in a high-impact journal. Thank you again for your valuable feedback and for considering our manuscript.

Response to Reviewer 4's Comments

Comments 1: Transitions shown in Figure 7a are barely legible and should be increased in size or moved to SI.

Response 1: Thank you for your valuable comment. We agree that the transitions displayed in the figure are indeed given in a too small font size. We have enlarged the size of the transitions in Figure 7a to make them more prominent and visible, and have updated the figure in the revised version of our manuscript.

Comments 2: The results on the challenges for scale-up should be mentioned in the main manuscript text.

Response 2: Thank you for your valuable comment. We have included a discussion of the challenges for scale-up in the main manuscript text. This discussion highlights the key challenges and implications for future research in this area. We believe this addition will further strengthen the paper.